# Abstain Mask Retain Core: Time Series Prediction by Adaptive Masking Loss with Representation Consistency

**Renzhao Liang**[*]
Beihang University
liangrenzhao@buaa.edu.cn

**Sizhe Xu**[*]
New York University
sx2490@nyu.edu

**Chenggang Xie**
Beihang University
xiechenggang@buaa.edu.cn

**Jingru Chen**
Peking University
2401212839@pku.edu.cn

**Feiyang Ren**[†]
New York University
fr2303@nyu.edu

**Shu Yang**
New York University
sy4254@nyu.edu

**Takahiro Yabe**[‡]
New York University
takahiroyabe@nyu.edu

## Abstract

Time series forecasting plays a pivotal role in critical domains such as energy management and financial markets. Although deep learning-based approaches (e.g., MLP, RNN, Transformer) have achieved remarkable progress, the prevailing "long-sequence information gain hypothesis" exhibits inherent limitations. Through systematic experimentation, this study reveals a counterintuitive phenomenon: appropriately truncating historical data can paradoxically enhance prediction accuracy, indicating that existing models learn substantial redundant features (e.g., noise or irrelevant fluctuations) during training, thereby compromising effective signal extraction. Building upon information bottleneck theory, we propose an innovative solution termed Adaptive Masking Loss with Representation Consistency (AMRC), which features two core components: 1) Dynamic masking loss, which adaptively identified highly discriminative temporal segments to guide gradient descent during model training; 2) Representation consistency constraint, which stabilized the mapping relationships among inputs, labels, and predictions. Experimental results demonstrate that AMRC effectively suppresses redundant feature learning while significantly improving model performance. This work not only challenges conventional assumptions in temporal modeling but also provides novel theoretical insights and methodological breakthroughs for developing efficient and robust forecasting models. We have made our code available at https://github.com/MazelTovy/AMRC.

## 1 Introduction

Time series forecasting, as a pivotal technology in critical domains such as energy management and financial markets, directly influences decision-making quality and economic efficiency [11, 13, 19, 20, 23]. Recent breakthroughs in deep learning have driven revolutionary advancements in

---

[*]Equal contribution.

[†]Now at University of Leeds.

[‡]Corresponding author.

39th Conference on Neural Information Processing Systems (NeurIPS 2025).

time series prediction. Contemporary frameworks including Multilayer Perceptron (MLP)-based architectures [4, 7, 18, 29, 30, 33], Recurrent Neural Networks (RNNs) with their variants [9, 14, 22], and attention mechanism-based models exemplified by the Transformer [2, 6, 17, 21, 36, 37, 39], have achieved remarkable breakthroughs in modeling complex temporal patterns through the construction of elaborate hierarchical temporal dependencies.

Current mainstream forecasting models predominantly adhere to the "long-sequence information gain hypothesis," which posits that extending historical data length enhances the availability of temporal dependencies [16, 34]. However, through systematic experimental analysis, this study challenges this conventional assumption. As shown in Table 1, we observed a counterintuitive phenomenon across multiple benchmark datasets and diverse model architectures: appropriately truncating early segments of input sequences can significantly improve prediction accuracy. This finding reveals a critical issue in modern predictive models: during training, models inadvertently capture a substantial number of redundant features. These features not only fail to enhance performance but also interfere with the learning process, thereby limiting the models' potential to achieve optimal results.

Through systematic analysis, we have identified two typical manifestations of redundant features and their underlying mechanisms. First, input truncation optimization experiments (as shown in Figure 2b and Table 1) demonstrate that selectively masking partial historical data can significantly improve model prediction performance. This phenomenon reveals the current model's inefficient utilization of long historical windows. Second, representation similarity analysis (as illustrated in Figure 2a) shows that both the model's prediction results and intermediate embeddings exhibit an abnormally concentrated distribution, which significantly deviates from the natural dispersion characteristics of the input and label. Collectively, these observations indicate that existing models exhibit low efficiency when processing long historical windows, often encoding substantial noise or irrelevant variables rather than truly predictive signals.

Building upon information bottleneck theory [10, 24, 26, 27], this study proposes an innovative method called Adaptive Masking Loss with Representation Consistency (AMRC). The core methodology comprises: 1) An adaptive masking mechanism that dynamically identifies key segments with high discriminative power in sequential data and leverages these informative segments to guide the gradient optimization process (as illustrated in Figure 3) ; 2) A representation consistency constraint that establishes stable mapping relationships among the input feature space, label space, and predicted outputs, thereby effectively enhancing the model's generalization capability. Experimental results (as shown in Table 2) demonstrate that the AMRC method significantly reduces the complexity of the training solution space by suppressing the model's reliance on redundant features, fully exploits the performance potential of the model architecture, and consequently improves prediction accuracy.

The primary contributions of this study include:

- **Theoretical Insight:** Through rigorous experimental validation, We demonstrate that existing time series forecasting models are prone to learning redundant features, which in turn constrain their performance. Building on the theory of information bottlenecks, we construct a novel theoretical framework for time series modeling and propose an innovative optimization pathway, offering a new theoretical perspective for advancing the field of time series forecasting.
- **Methodological Innovation:** We propose an optimization framework Adaptive Masking Loss with Representation Consistency. By dynamically selecting discriminative temporal segments to guide gradient descent (as illustrated in Figure 1) while enforcing input-label-prediction consistency, our method effectively suppresses redundant feature learning. Extensive experiments demonstrate consistent performance gains across diverse benchmarks and architectures.

Our work advances the understanding of temporal pattern learning mechanisms while offering a practical pathway to enhance the efficiency and reliability of time series forecasting systems.

## 2 Related Work

The Information Bottleneck (IB) method was first introduced by Tishby et al. [26] as an information-theoretic framework that aims to compress input signals while preserving as much relevant information as possible about the target output. In the field of machine learning, IB theory has been widely adopted as a regularization technique. For instance, Alemi et al. [1] proposed the Variational Information Bottleneck (VIB), which leverages variational inference to construct a tractable lower bound on the

IB objective. Building upon this, Tishby and Zaslavsky [27]further explored the applicability of information-theoretic objectives to deep neural networks. Research on IB has also extended into the domain of clustering. Slonim et al. [24] developed a distributional clustering algorithm based on mutual information maximization and demonstrated its effectiveness on the 20 Newsgroups dataset, achieving substantial compression with minimal loss of relevant information. More recently, Hu et al. [10] conducted a comprehensive survey of the IB literature, reviewing over two decades of theoretical developments, methodological advances, and practical applications.

In the context of deep learning, time series forecasting methods can be broadly categorized into MLP, RNN, and Transformer-based approaches. Among MLP-based models, DLinear [33] and TSMixer [7] are representative examples, featuring relatively simple architectures while achieving strong performance across multiple datasets. RNN-based methods, such as Segrnn [6] and LSTMlong [22], focus on structural modifications to address challenges related to parallel prediction and long-sequence modeling. Transformer-based models include Informer [37], Autoformer [21], and iTransformer [15]. Informer introduces a sparse attention mechanism to improve the scalability of traditional attention for time series modeling; Autoformer incorporates frequency-domain information to enhance attention; and iTransformer further extends attention across channels by embedding multivariate sequences for variable-aware representation.

Another key research area concerns noise robustness and representation learning. Early work, such as Informer [37], used sparse attention for information distillation in long sequences, while TS2Vec [31] adopted contrastive learning to regularize temporal representations. More recently, dedicated frameworks have been proposed. For instance, TS-CoT [35] employs a dual-encoder architecture and a cross-view prototype alignment mechanism to achieve global semantic consistency. Similarly, DECL [38] guides contrastive learning to acquire denoising capabilities by constructing positive samples from denoised data and leveraging an adaptive denoiser.

## 3   Analysis of Redundant Feature Learning

Given a multivariate time series $\mathbf{X} \in \mathbb{R}^{T \times D}$, where $T$ is the number of timesteps and $D$ is the number of variables, the objective of time series forecasting is to learn a mapping function $f_\theta$ that transforms historical observations $\mathbf{X}_{t-L:t} \in \mathbb{R}^{L \times D}$ (where $L$ denotes the input length ) into future values $\mathbf{X}_{t+1:t+H} \in \mathbb{R}^{H \times D}$ (where $H$ represents the forecasting horizon).

Conventional time series forecasting models follow the long-sequence information gain hypothesis [3, 5, 32, 37], which holds that increasing the input length $L$ improves forecasting accuracy. However, our experiments (Table 1) on multiple standard benchmarks reveal a counterintuitive result: truncating the input—such as masking the first $k$ timesteps—often improves forecasting performance, which is measured by Mean Squared Error (MSE). We found that models tend to learn redundant features, which degrade model performance even after convergence. This finding is supported by two key observations:

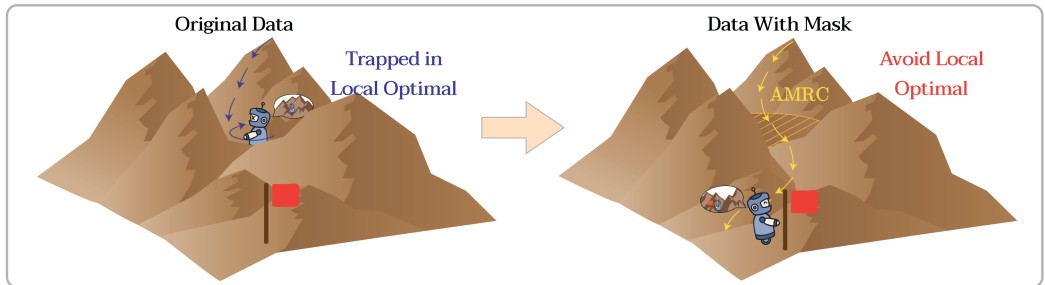

Figure 1: Illustration of the effect of AMRC method. Without regularization, the model tends to overfit redundant input features, leading to suboptimal convergence. By suppressing redundant input features, AMRC restructures the optimization landscape, promoting more efficient representation learning and facilitating better convergence.

## 3.1 Input Truncation Optimization

Based on the baseline model configuration (input length $L = 48$, forecasting horizon $H = 48$), we design an input truncation comparative experiment by applying a masking operator $\mathcal{M}_k(\cdot)$ to the input sequence. When we have an input sequence of length $L$ at time step $t$, denoted as $\boldsymbol{X}_t^{(L)}$, the masking operator $\mathcal{M}_k(\cdot)$ is mathematically defined as:

$$\mathcal{M}_k(\boldsymbol{X}_t^{(L)}) = \begin{cases} 0 & \text{if } i \leq k \\ \boldsymbol{X}_t^{(L)} & \text{otherwise} \end{cases} \tag{1}$$

Here, $k \in \{1, \ldots, L\}$ denotes the masking step size.

To probe redundant features, we employ an Optimal Masking strategy: Given an input sequence of length $L$, we generate $L$ masked variants $\{\mathcal{M}_k(\boldsymbol{X}_t^{(L)})\}_{k=1}^L$ (zero-padded to preserve dimensionality). For instance, $k = 5$ yields $L' = 43$ (first 5 positions zeroed). The optimal mask length $k^*$ is selected as the configuration minimizing MSE, thereby defining the theoretical upper bound for redundancy elimination:

$$k^* = \underset{k \in \{1,2,\ldots,L\}}{\arg\min} \ \mathbb{E}\left[\left\| f_\theta\big(\mathcal{M}_k(\boldsymbol{X}_t^{(L)})\big) - \boldsymbol{Y}_t^{(H)} \right\|^2\right] \tag{2}$$

Table 1: Performance Gains via Optimal Masking Across Time Series Models. Ratio quantifies the percentage of training samples demonstrating prediction error reduction through Optimal Masking, calculated as *number of masked series/number of total series* $\times 100\%$

| Model | | ETTh1 | | | ETTh2 | | | Solar-Energy | | | Weather | | |
|---|---|---|---|---|---|---|---|---|---|---|---|---|---|
| Metric | | MSE | MSE* | Ratio | MSE | MSE* | Ratio | MSE | MSE* | Ratio | MSE | MSE* | Ratio |
| SOFTS | Train Set | 0.278 | **0.254** | 56.54% | 0.318 | **0.259** | 61.65% | 0.182 | **0.155** | 11.80% | 0.421 | **0.400** | 45.10% |
| | Test Set | 0.408 | **0.365** | 64.24% | 0.326 | **0.303** | 28.73% | 0.293 | **0.184** | 41.58% | 0.205 | **0.185** | 54.93% |
| iTransformer | Train Set | 0.298 | **0.270** | 57.87% | 0.315 | **0.261** | 64.19% | 0.410 | **0.281** | 61.97% | 0.436 | **0.389** | 62.98% |
| | Test Set | 0.413 | **0.289** | 60.07% | 0.329 | **0.299** | 32.16% | 0.395 | **0.271** | 68.43% | 0.209 | **0.170** | 80.26% |
| PatchTST | Train Set | 0.343 | **0.303** | 65.57% | 0.329 | **0.269** | 69.35% | 0.366 | **0.277** | 35.89% | 0.227 | **0.180** | 45.55% |
| | Test Set | 0.424 | **0.402** | 65.51% | 0.327 | **0.298** | 42.46% | 0.374 | **0.344** | 51.66% | 0.215 | **0.180** | 42.43% |
| TSMixer | Train Set | 0.372 | **0.342** | 55.79% | 0.544 | **0.431** | 73.96% | 0.233 | **0.195** | 26.30% | 0.363 | **0.348** | 37.57% |
| | Test Set | 0.402 | **0.372** | 59.19% | 0.324 | **0.289** | 42.13% | 0.288 | **0.250** | 40.12% | 0.222 | **0.195** | 70.88% |
| TimeMixer | Train Set | 0.290 | **0.262** | 57.96% | 0.309 | **0.251** | 59.36% | 0.142 | **0.112** | 13.58% | 0.403 | **0.353** | 63.93% |
| | Test Set | 0.393 | **0.366** | 58.04% | 0.318 | **0.285** | 44.52% | 0.288 | **0.253** | 36.25% | 0.197 | **0.172** | 66.13% |

As demonstrated in Table 1, the experimental results confirm that masked models consistently achieve lower MSE, with more than 50% of samples exhibiting improved predictive performance (Ratio > 50%). Notably, the phenomenon of redundancy learning shows strong architecture-agnostic characteristics. On the Weather dataset, both iTransformer (a Transformer-based model) and TSMixer (an MLP-based model) demonstrate similar relative improvements: iTransformer achieves an MSE reduction from **0.209** to **0.170** ($-18.7\%$), while TSMixer improves from **0.222** to **0.195** ($-12.2\%$). These results indicate that the effectiveness of our masking strategy is not dependent on specific model architectures.

## 3.2 Representation Similarity Paradox

To further investigate the redundant feature learning phenomenon, we apply t-SNE to project the SOFTS model's high-dimensional representations of the input, embedding, prediction, and label onto a 2D plane (Figure. 2a), after normalizing all features to the $[0, 1]$ range.

As illustrated in Figure. 2a, Normalized input ($\mathbf{Z}_{\text{in}} \in \mathbb{R}^L$) and output ($\mathbf{Z}_{\text{out}} \in \mathbb{R}^H$) embeddings show a clear contrast: inputs remain dispersed, while embeddings and preds cluster tightly despite large differences in their corresponding labels. This suggests that the model encodes redundant, task-irrelevant features that misrepresent semantic relationships and distort the input-output mapping.

## 3.3 Information Bottleneck Constraints on Redundancy

In time-series forcasting models, the input sequence $X$ is typically encoded into a latent representation $Z$, from which a decoder then predicts the target sequence $Y$. The optimization objective is to learn

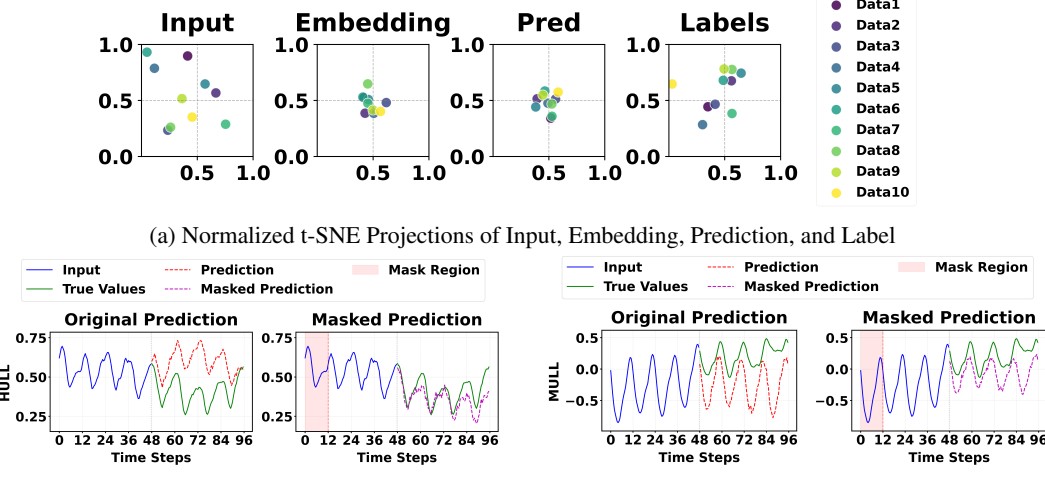

(a) Normalized t-SNE Projections of Input, Embedding, Prediction, and Label

(b) Masked vs. Unmasked Prediction Performance

Figure 2: Embedding Distributions and Masking Effects of Our Method.

an optimal representation $Z$ that maximally preserves information relevant to $Y$ while discarding irrelevant details from $X$. According to the Information Bottleneck (IB) Theory [24], this process can be viewed as a bottleneck that compresses input information. The informational relationships among $X$, $Y$, and $Z$, which are governed by the model's learnable parameter $\theta$, can be quantified using mutual information. The objective can be thus formally expressed as maximizing the mutual information between the representation $Z$ and the target $Y$:

$$I(Z, Y; \boldsymbol{\theta}) = \int dx\, dy\, p(z, y \mid \boldsymbol{\theta}) \log \frac{p(z, y \mid \boldsymbol{\theta})}{p(z \mid \boldsymbol{\theta}) p(y \mid \boldsymbol{\theta})}. \tag{3}$$

Due to inherent limitations in the data and model capacity, the amount of information that can be extracted and transmitted during training is bounded. Consequently, the representation capacity is subject to an upper information constraint $I_c$. Based on this, the objective of the time series prediction model can be equivalently formulated as the following constrained optimization problem:

$$\max_{\boldsymbol{\theta}} I(Z, Y; \boldsymbol{\theta}) \quad \text{s.t.} \quad I(X, Z; \boldsymbol{\theta}) \leq I_c. \tag{4}$$

This constrained optimization problem can be transformed into an unconstrained form using the method of Lagrange multipliers, leading to the maximization of the following objective [1]:

$$R_{\mathrm{IB}}(\boldsymbol{\theta}) = I(Z; Y; \boldsymbol{\theta}) - \beta I(Z; X; \boldsymbol{\theta}). \tag{5}$$

There are two implementation paths under this objective: one is to maximize the mutual information $I(Z; Y)$ between $Z$ and $Y$; the other is to minimize the mutual information $I(Z; X)$ between $Z$ and $X$. Most current sequential prediction models focus on improving $I(Z; Y)$ through iterative training, but have not explicitly optimized performance by penalizing redundant features via minimizing $I(Z; X)$. Therefore, we propose an adaptive loss function that aims to minimize the mutual information between $X$ and $Z$, offering a novel optimization path for improving the performance of sequential prediction models.

## 4 Proposed Method

### 4.1 Adaptive Masking Loss (AML)

As discussed in Section 3.1, applying ideal masking to input data reduces the information $I(X)$ while improving prediction accuracy. This indicates that the representation $Z_{k^*}$, generated by encoder $p_\theta$ from masked features $X_{t,k^*}$, contains less redundancy and better approximates the minimal sufficient statistics (i.e., with smaller $I(X, Z_{k^*}; \theta)$). Based on this insight, we propose the **Adaptive Masking Loss (AML)** to explicitly reduce mutual information $I(X, Z; \theta)$ by guiding the encoder's output representation $Z$ toward $Z_{k^*}$, thereby suppressing redundant feature learning and unleashing model potential. The overall framework of AML is illustrated in Figure 3.

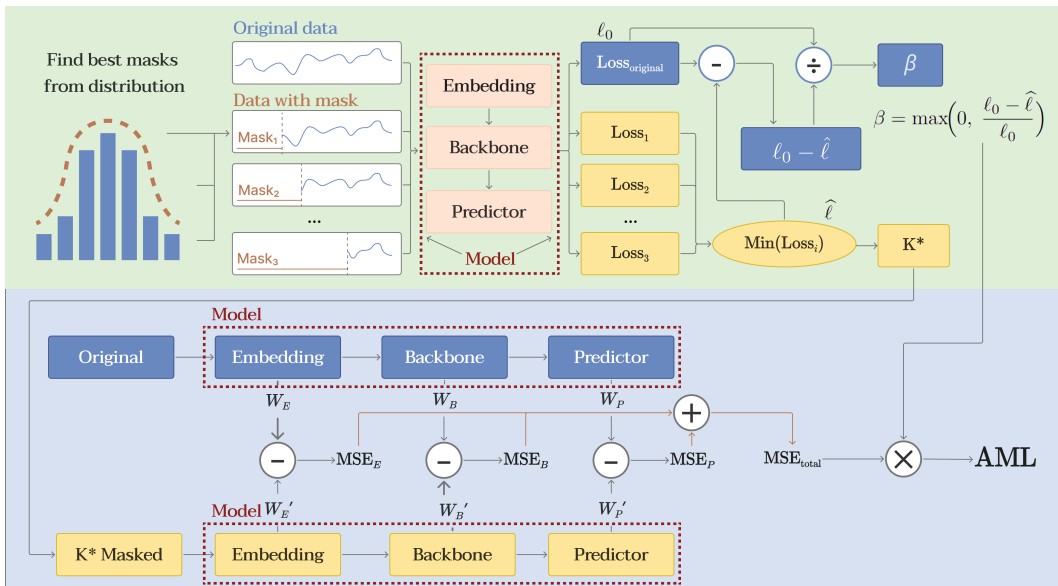

Figure 3: Overview of the Adaptive Masking Loss (AML) framework. The upper half illustrates how the optimal mask length $K^*$ is selected by evaluating prediction losses over sampled masks. A weighting coefficient $\beta$ is computed based on the gain over the unmasked loss. The lower half shows the AML loss, calculated as the sum of representation differences between the original input and the $K^*$ masked input across embedding, backbone, and predictor layers.

#### 4.1.1 Implementation

The exhaustive search for optimal mask $k^*$ by enumerating all possible mask lengths $k \in \{1, ..., L\}$ results in prohibitive $O(L)$ time complexity for long sequences. We therefore adopt an efficient stochastic approximation strategy:

1. **Random Mask Generation**: Independently sample $m$ mask indices $\{k_s\}_{s=1}^m$ from uniform distribution $d(k) = \text{Uniform}\{1, ..., L\}$, each generating a masked variant:

$$\widetilde{X}_{t,s}^{(L)} = \mathcal{M}_{k_s}(X_t^{(L)}) \tag{6}$$

2. **Loss Evaluation**: Compute prediction losses for both masked and original data:

$$\ell_s = \mathcal{L}(f_\theta(\widetilde{X}_{t,s}^{(L)}), Y_t^{(H)}) \tag{7}$$

$$\ell = \mathcal{L}(f_\theta(X_t^{(L)}), Y_t^{(H)}) \tag{8}$$

3. **Optimal Representation Selection**: If $\exists \ell_s < \ell$, the corresponding representation $\widetilde{Z}_s = p_\theta(\widetilde{X}_{t,s}^{(L)})$ satisfies $I(X_t^{(L)}, \widetilde{Z}_s) \le I(X_t^{(L)}, Z)$, where $Z = p_\theta(X_t^{(L)})$ is the original representation. It signifies that a masked input variant can achieve better predictive performance than the original input. This provides a clear indication that the removed information was redundant rather than essential. The optimal mask variant is selected by:

$$s^* = \arg\max_s(\ell - \ell_s) \tag{9}$$

#### 4.1.2 Loss Formulation

To promote compact and informative representations, AML minimizes the distance between the original representation $Z$ and the optimal masked variant $\widetilde{Z}_{s^*}$:

$$\mathcal{L}_{\text{AML}} = \beta \cdot \frac{1}{D_1 \times D_2} \|Z - \widetilde{Z}_{s^*}\|^2 \tag{10}$$

The adaptive weight $\beta = \max(0, (\ell - \ell_{s^*})/\ell)$ ensures that this regularization term is only active when a better-performing masked representation is found. Such a setup dynamically scales the optimization intensity, guaranteeing a more substantial influence from mask variants with greater loss reduction.

## 4.2 Embedding Similarity Penalty (ESP)

Time series forecasting models often encounter two issues: semantic inconsistency, where semantically similar inputs lead to substantially different predictions, and representation collapse, where dissimilar inputs result in nearly identical outputs. While consistency regularization methods like Temporal Ensembling [12] and Mean Teacher [25] address stability for individual samples under augmentation, they do not explicitly consider the relational structure between different samples. We therefore introduce the Embedding Similarity Penalty (ESP), a strategy that directly addresses this by comparing the geometry of the embedding space with that of the output space for pairs of samples within a mini-batch.

**Pairwise distances.** For a batch $\mathcal{B} = \{(X_i, Y_i)\}_{i=1}^n$ we denote by $Z_i = f_{\text{enc}}(X_i) \in \mathbb{R}^{L \times D}$ the encoder output and keep the ground-truth $Y_i \in \mathbb{R}^{P \times D}$. The (normalised) squared Frobenius distances are

$$\Delta_{ij}^E = \frac{1}{L \times D} \|Z_i - Z_j\|_F^2, \qquad \Delta_{ij}^O = \frac{1}{P \times D} \|Y_i - Y_j\|_F^2, \quad 1 \le i, j \le n. \tag{11}$$

**Consistency penalty.** Ideally $\Delta_{ij}^E$ and $\Delta_{ij}^O$ should match: semantically similar inputs ($\Delta_{ij}^E \approx 0$) ought to produce similar outputs ($\Delta_{ij}^O \approx 0$), and vice versa. Deviation is quantified element-wise through

$$P_{ij} = \text{ReLU}\big(\Delta_{ij}^E - \Delta_{ij}^O\big) + \text{ReLU}\big(\Delta_{ij}^O - \Delta_{ij}^E\big) = |\Delta_{ij}^E - \Delta_{ij}^O|_+, \tag{12}$$

where $\text{ReLU}(x) = \max(0, x)$ and $|\cdot|_+$ denotes the non-negative part. The **Embedding-Similarity Penalty** then reads

$$\mathcal{L}_{\text{ESP}} = \frac{1}{n^2} \sum_{i=1}^n \sum_{j=1}^n P_{ij}. \tag{13}$$

Equation (13) back-propagates smooth, unbiased gradients that jointly reshape the encoder and the predictor so that input and output manifolds remain geometrically aligned. The detailed implementation of the Embedding Similarity Penalty (ESP) is provided as pseudocode in Appendix C Algorithm 1.

## 4.3 Overall Training Objective

Section 4.1 introduced the Adaptive Masking Loss $\mathcal{L}_{\text{AML}}$ that discourages the learning of redundant temporal prefixes, while Section 4.2 proposed the Embedding-Similarity Penalty $\mathcal{L}_{\text{ESP}}$ to enforce semantic–behavioural consistency. Combined with the standard prediction loss $\mathcal{L}_{\text{pred}}$ (*e.g.*, MSE between the forecast $\hat{Y}$ and the target $Y$), our final objective is

$$\mathcal{L}_{\text{total}} = \mathcal{L}_{\text{pred}} + \lambda_{\text{AML}} \mathcal{L}_{\text{AML}} + \lambda_{\text{ESP}} \mathcal{L}_{\text{ESP}}, \tag{14}$$

where $\lambda_{\text{AML}}, \lambda_{\text{ESP}} > 0$ control the strength of each auxiliary term. Minimizing (14) jointly (i) identifies the informative prefix for every sequence, (ii) preserves the intrinsic topology of the data, and (iii) improves predictive accuracy and interpretability without adding inference-time overhead.

# 5 Experiment

## 5.1 Experiment Setup

**Datasets.** We evaluate our proposed method using seven widely recognized benchmark datasets for multivariate time series forecasting: **ETTh1**, **ETTh2**, **ETTm1**, **ETTm2**, **Solar-Energy**, **Electricity**, and **Weather**. These datasets encompass a variety of application scenarios with different temporal resolutions, seasonality patterns, and dynamic structures. Detailed descriptions of each dataset, including their specific characteristics and collection periods, are provided in the Appendix E.

**Task formulation.** In our experimental setup, the forecasting task is formulated as a sequence-to-sequence regression problem, applicable to multivariate time series. Each model is trained to predict a future sequence $\boldsymbol{Y}_t^{(H)} \in \mathbb{R}^{H \times D}$ from a fixed-length historical input sequence $\boldsymbol{X}_t^{(48)} \in \mathbb{R}^{48 \times D}$,

where $H$ denotes the prediction length and $D$ is the number of variables. We adopt multiple prediction horizons $H \in \{48, 72, 96, 120, 144, 168, 192\}$.

**Baselines.** Our method is compared against five diverse baseline models: **SOFTS** [8], **iTransformer** [15], **PatchTST** [17], **TSMixer** [7], and **TimeMixer** [28]. These baselines are implemented using their official codebases and recommended hyperparameters to ensure a fair comparison under consistent experimental conditions.

**Implementation details.** All models are implemented in PyTorch and trained on a single NVIDIA A100 80GB GPU. To ensure a fair comparison and allow both baseline models and those augmented with our proposed modules to fully exploit their capacity, we train each model for up to 100 epochs using the Adam optimizer with an initial learning rate of $1 \times 10^{-4}$, a cosine annealing scheduler, and a batch size of 32. Early stopping is applied based on validation loss with a patience of 20 epochs. The best-performing checkpoint on the validation set is selected for final evaluation on the test set.

**Hyperparameter selection.** For the AML, the input sequence prefix length is configured as $L = 48$, with the mask sampling cardinality parameterized as $m = 12$. We fix both $\lambda_{\text{AML}}$ and $\lambda_{\text{ESP}}$ to 1 for all experiments. These settings follow standard benchmark configurations commonly used in time series forecasting.

## 5.2 Forecasting Results

We present the forecasting performance of our method—Adaptive Masking Loss with Representation Consistency (AMRC)—in comparison with five representative baseline models across seven widely used time series benchmark datasets. Table 2 reports the Mean Squared Error (MSE) and Mean Absolute Error (MAE) for each model, both with and without the incorporation of AMRC.

Table 2: Performance Comparison of Time Series Forecasting Models With and Without AMRC. In the experimental results, we highlighted in bold the parts where the AMRC model improved by more than 0.005 in MSE and MAE metrics compared to the baseline model. The detailed hyperparameter configurations for each model can be found in Appendix B. Full results are listed in Appendix D.1 Table 5. Furthermore, a detailed statistical analysis presenting results as mean $\pm$ standard deviation over 10 runs, along with significance tests, is provided in Appendix D.1 Table 6. To further validate the robustness of AMRC, we conducted additional experiments on the ExchangeRate dataset and the challenging, low-data Illness dataset, as detailed in Appendix 7.

| Model | | ETTh1 | | ETTh2 | | ETTm1 | | ETTm2 | | Solar-Energy | | Electricity | | Weather | |
|---|---|---|---|---|---|---|---|---|---|---|---|---|---|---|---|
| Metric | | MSE | MAE | MSE | MAE | MSE | MAE | MSE | MAE | MSE | MAE | MSE | MAE | MSE | MAE |
| SOFTS | Original | 0.408 | 0.414 | 0.326 | 0.359 | 0.484 | 0.434 | 0.210 | 0.285 | 0.293 | 0.314 | 0.169 | 0.255 | 0.205 | 0.234 |
| | AMRC | **0.389** | **0.393** | **0.311** | 0.362 | **0.475** | **0.423** | **0.198** | **0.265** | 0.290 | 0.309 | **0.162** | **0.244** | **0.196** | **0.220** |
| iTransformer | Original | 0.413 | 0.415 | 0.329 | 0.362 | 0.517 | 0.448 | 0.213 | 0.290 | 0.395 | 0.352 | 0.176 | 0.260 | 0.209 | 0.237 |
| | AMRC | **0.402** | **0.399** | 0.324 | **0.356** | **0.502** | **0.447** | 0.211 | **0.280** | 0.392 | **0.342** | **0.163** | **0.239** | **0.201** | **0.221** |
| TimeMixer | Original | 0.393 | 0.408 | 0.318 | 0.355 | 0.466 | 0.429 | 0.209 | 0.285 | 0.288 | 0.317 | 0.194 | 0.279 | 0.197 | 0.237 |
| | AMRC | 0.388 | **0.401** | 0.316 | **0.339** | **0.447** | **0.405** | 0.204 | **0.269** | 0.284 | 0.317 | **0.188** | 0.277 | **0.186** | **0.228** |
| PatchTST | Original | 0.424 | 0.424 | 0.327 | 0.358 | 0.461 | 0.422 | 0.211 | 0.287 | 0.374 | 0.382 | 0.211 | 0.283 | 0.215 | 0.280 |
| | AMRC | **0.411** | **0.415** | **0.319** | 0.356 | 0.456 | **0.413** | **0.196** | **0.271** | **0.361** | 0.376 | 0.207 | 0.285 | 0.210 | **0.264** |
| TSMixer | Original | 0.402 | 0.412 | 0.324 | 0.357 | 0.440 | 0.413 | 0.201 | 0.279 | 0.288 | 0.314 | 0.172 | 0.258 | 0.222 | 0.288 |
| | AMRC | **0.386** | **0.397** | 0.319 | **0.340** | **0.432** | 0.412 | 0.196 | **0.257** | **0.280** | 0.313 | 0.169 | **0.247** | **0.212** | **0.281** |

**Consistent Performance Gains.** Across all models and datasets, our method consistently yields performance improvements. For example, the MSE of the SOFTS model decreases from 0.408 to 0.389 on the ETTh1 dataset. Similar trends are observed in iTransformer, where the MSE on Electricity drops from 0.176 to 0.163. The enhancements demonstrate that AMRC effectively mitigates redundant or noisy temporal segments, thereby improving prediction stability and accuracy.

**Architecture-Agnostic Effectiveness.** AMRC delivers significant performance gains not only on Transformer-based architectures such as iTransformer and PatchTST, but also on MLP-based models including TimeMixer, SOFTS, and TSMixer. For instance, on the ETTm2 dataset, the MSE of PatchTST model decreases from 0.211 to 0.196 (a reduction of approximately 7.11%), while the MSE of SOFTS model drops from 0.210 to 0.198 (approximately 5.71% reduction). These results demonstrate the strong architecture-agnostic generalization ability of AMRC, highlighting its broad applicability across a wide range of time series forecasting models.

**Generalization on Low-Channel Datasets.** On datasets with fewer input channels (ETTh1, ETTh2, ETTm1, ETTm2), AMRC effectively enhances model performance. For instance, on ETTm1, the MSE of iTransformer decreases from 0.517 to 0.502, and that of TSMixer drops from 0.440 to 0.432. These results demonstrate AMRC's ability to mitigate overfitting and improve prediction accuracy in low-dimensional time series forecasting tasks.

**Robustness on High-Channel Datasets.** For high-dimensional datasets such as Weather (21 channels) and Solar-Energy (137 channels) see in Appendix E, AMRC consistently improves robustness by reducing the impact of signal noise and inter-channel redundancy. On the Weather dataset, TimeMixer's MSE decreases from 0.197 to 0.186 and MAE from 0.237 to 0.228, while iTransformer sees an MAE drop from 0.237 to 0.221. On Solar-Energy, PatchTST's MSE drops from 0.374 to 0.361, and SOFTS sees a slight MAE reduction from 0.314 to 0.309. These enhancements highlight AMRC's effectiveness in managing complexity in multivariate time series with high channel counts.

**Generalizable Training Framework.** The consistent performance improvements observed across all models validate the strong scalability and integrability of AMRC. As a constraint-based optimization strategy, AMRC does not rely on any specific model architecture, making it highly generalizable. It serves as a versatile training framework for enhancing both the efficiency and accuracy of time series forecasting models.

## 5.3 Ablation Study

**Setup.** We evaluate ablation variants on four diverse datasets: ETTh1 and ETTh2, representing hourly electricity load with varying degrees of seasonality; Solar-Energy, which exhibits weather-driven variability and periodicity; and Weather, a multivariate meteorological dataset with complex inter-variable dependencies. We adopt a fixed input horizon following standard benchmarks. We also analyzed the sensitivity to the number of sampled masks, $m$, used in AML. While a larger $m$ allows for a more extensive search, it incurs greater computational cost. Our analysis, detailed in Appendix Table 8, reveals diminishing returns as $m$ increases. Consequently, we set $m = 12$ for all experiments to effectively balance performance and computational efficiency.

**Evaluation protocol.** For each dataset, we apply the ablation study to five baseline models SOFTS, iTransformer, TimeMixer, PatchTST, and TSMixer under four configurations:1) baseline + AML, 2) baseline + ESP, and 3) baseline + both AML and ESP. This design allows us to assess the standalone effectiveness of each module as well as their combined synergy.

**Findings.** We evaluate the individual and joint effects of the AML and ESP components using five representative forecasting architectures across four datasets. As shown in Table 3, both components contribute measurable performance gains in isolation, while their combination AMRC consistently leads to the best forecasting accuracy in terms of MSE and MAE. AML provides stronger improvements across most settings, supporting its role in suppressing redundant prefixes during training. ESP, while often delivering smaller standalone gains, remains beneficial by promoting geometric alignment between embedding and output spaces. Together, these findings demonstrate that each component addresses a distinct source of generalization error.

**Component impact across architectures.** The benefits of AML and ESP are consistently observed across all backbone models, regardless of architectural differences. For instance, models with strong expressiveness, such as iTransformer and TimeMixer, benefit significantly from AML, achieving notable MSE reductions on datasets like Weather and ETTh2. Even architectures without attention mechanisms, such as SOFTS and TSMixer, exhibit consistent gains, highlighting the broad applicability of adaptive prefix masking. In contrast, the improvements from ESP are often more dataset-dependent, being particularly effective on high-dimensional multivariate inputs where representation alignment plays a critical role. For example, ESP yields non-trivial reductions in MAE on Weather, where multiple variables evolve under shared dynamics. Notably, we observe relatively smaller improvements on the Solar-Energy dataset for transformer-based models such as PatchTST and iTransformer, which may be attributed to their reliance on longer input sequences for stable attention computation.

**Complementarity and synergy.** The AMRC configuration, which jointly applies AML and ESP, consistently outperforms its ablated variants across all benchmarks. The performance improvement

Table 3: Ablation Study Results on Different Model Components

| Model | Metric | ETTh1 | | ETTh2 | | Solar-Energy | | Weather | |
|---|---|---|---|---|---|---|---|---|---|
| | | MSE | MAE | MSE | MAE | MSE | MAE | MSE | MAE |
| SOFTS | AML only | 0.401 | 0.405 | 0.322 | 0.358 | 0.297 | 0.309 | 0.192 | 0.228 |
| | ESP only | 0.393 | 0.398 | 0.318 | 0.351 | 0.295 | 0.318 | 0.208 | 0.241 |
| | AMRC | **0.389** | **0.393** | **0.311** | **0.362** | **0.290** | **0.309** | **0.196** | **0.220** |
| iTransformer | AML only | 0.410 | 0.413 | 0.328 | 0.363 | 0.398 | 0.347 | 0.205 | 0.230 |
| | ESP only | 0.407 | 0.408 | 0.326 | 0.359 | 0.402 | 0.351 | 0.210 | 0.248 |
| | AMRC | **0.402** | **0.399** | **0.324** | **0.356** | **0.392** | **0.342** | **0.201** | **0.221** |
| TimeMixer | AML only | 0.395 | 0.412 | 0.319 | 0.351 | 0.287 | 0.319 | 0.189 | 0.232 |
| | ESP only | 0.391 | 0.406 | 0.317 | 0.347 | 0.293 | 0.325 | 0.202 | 0.248 |
| | AMRC | **0.388** | **0.401** | **0.316** | **0.339** | **0.284** | **0.317** | **0.186** | **0.228** |
| PatchTST | AML only | 0.419 | 0.420 | 0.325 | 0.361 | 0.369 | 0.379 | 0.214 | 0.274 |
| | ESP only | 0.417 | 0.418 | 0.323 | 0.357 | 0.375 | 0.384 | 0.217 | 0.281 |
| | AMRC | **0.411** | **0.415** | **0.319** | **0.356** | **0.361** | **0.376** | **0.210** | **0.264** |
| TSMixer | AML only | 0.396 | 0.404 | 0.324 | 0.356 | 0.285 | 0.317 | 0.216 | 0.283 |
| | ESP only | 0.390 | 0.399 | 0.322 | 0.352 | 0.291 | 0.323 | 0.224 | 0.292 |
| | AMRC | **0.386** | **0.397** | **0.319** | **0.340** | **0.280** | **0.313** | **0.212** | **0.281** |

from combining both components generally exceeds the stronger of the two individual effects, indicating synergistic interaction. This complementarity can be attributed to their distinct operational scopes: AML operates on the input level by learning to suppress non-informative temporal segments, while ESP regularizes the latent space to align representations across semantically related inputs. As a result, AMRC improves both the quality of features learned from the data and the consistency of their usage in prediction. The robust gains observed across datasets and architectures suggest that jointly addressing input redundancy and representation inconsistency is critical for improving generalization in time series forecasting.

Table 4: AMRC Effectiveness with Prefix Masking at a Fixed Input Length ($L = 48$). Ratio is the percentage of training samples with reduced MSE under prefix masking. Ratio* is the same metric after training with AMRC. Average results across all lengths are in Appendix D.1 Table 10.

| Model | ETTh1 | | ETTh2 | | Solar-Energy | | Weather | |
|---|---|---|---|---|---|---|---|---|
| Metric | Ratio | Ratio* | Ratio | Ratio* | Ratio | Ratio* | Ratio | Ratio* |
| SOFTS | 64% | **57.33%** | 28.72% | **20.28%** | 41.58% | **33.49%** | 54.93% | **47.12%** |
| iTransformer | 60.07% | **49.95%** | 32.16% | **23.28%** | 68.43% | **63.21%** | 80.26% | **70.29%** |
| TimeMixer | 58.04% | **46.29%** | 44.52% | **34.17%** | 36.25% | **27.90%** | 66.13% | **52.28%** |
| PatchTST | 65.51% | **51.63%** | 42.46% | **26.19%** | 51.66% | **47.64%** | 42.43% | **30.78%** |
| TSMixer | 59.19% | **46.62%** | 42.13% | **27.98%** | 40.12% | **28.36%** | 70.88% | **58.23%** |

**Effectiveness of AMRC in Reducing Redundant Features** We evaluate the model's robustness to redundant input by computing the proportion of training samples with improved MSE under prefix masking Ratio and compare it to the value after applying AMRC Ratio*. As shown in Table 4, AMRC consistently improves or maintains this ratio, indicating its effectiveness in suppressing the impact of redundant temporal information.

## 6 Conclusion

This study pioneers the investigation into the negative effects of redundant feature learning in time series forecasting and introduces AMRC, a plug-and-play solution that suppresses such learning without requiring architectural modifications. Unlike prior work focused on enhancing predictive features, AMRC improves accuracy by reducing reliance on redundant features while maintaining model flexibility. Its key advantages include: 1) seamless integration with existing models, 2) effective suppression of feature redundancy, and 3) strong generalization performance across benchmark tests. By addressing the long-overlooked issue of redundant learning, this research provides a novel and practical methodology for optimizing forecasting models.

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

# A Limitations

Despite the demonstrated effectiveness (Table 2) of our approach, AMRC has several limitations related to its underlying assumptions, interpretability, and practical trade-offs, which highlight important directions for future work.

1. Limitations of AML AML's efficacy is bound by two key factors: the temporal characteristics of the data and the interpretability of its masking mechanism.

   - The prefix-masking strategy assumes that redundant information often resides in the initial segments of a time series. In scenarios where the most critical predictive information lies exclusively in the later portions of the input sequence, AML's core mechanism becomes ineffective. Masking the prefix will not improve the prediction loss, causing the adaptive coefficient $\beta$ to remain zero and deactivating the regularization.

   - A significant limitation is the "black box" nature of the masking process. While AML is designed to identify and suppress redundancy, it is difficult to determine precisely what kinds of patterns are being masked—whether they represent noise, outliers, or simply outdated information. The adaptive-weight mechanism improves efficiency, but the decision process is not transparent. Clarifying this is a crucial direction for future work to enhance the method's interpretability.

2. Dependency on Data Dimensionality and the Role of ESP

   - We observe that ESP's improvements are more pronounced on datasets with lower feature dimensionality (e.g., the ETTh family). On higher-dimensional datasets like Weather (21 channels) and Solar-Energy (137 channels), its standalone gains are comparatively smaller.

   - This occurs because ESP aligns the geometric structure between the embedding and output spaces. As feature dimensionality increases, the optimization directions for this alignment grow exponentially, introducing greater uncertainty during training and potentially yielding diminished returns.

   - This limitation is effectively mitigated within the combined AMRC framework. High-dimensional datasets often contain significant feature redundancy, which is precisely the condition where AML excels. Therefore, the two components are highly complementary: ESP is most effective in lower-dimensional settings, while AML provides the primary benefit in higher-dimensional, redundant settings, ensuring that AMRC remains robust across diverse data types.

3. Inherent Design Trade-offs The search for an optimal mask requires evaluating $m$ variants per batch, increasing the training cost by a factor of approximately $m$. This makes it less suitable for latency-sensitive applications.
   The optimal mask is found via stochastic sampling of $m$ candidates, which is an approximation of an exhaustive search. This practical compromise means that some redundancy may remain, though it strikes a balance with computational feasibility.

# B Details of the Baseline Model

All models are reproduced based on their official open-source implementations:

1. **SOFTS** from `https://github.com/Secilia-Cxy/SOFTS`.

2. **TimeMixer** from `https://github.com/kwuking/TimeMixer`.

3. **iTransformer** from `https://github.com/thuml/iTransformer`.

4. **PatchTST** from `https://github.com/yuqinie98/PatchTST`.

5. **TSMixer** from `https://github.com/ditschuk/pytorch-tsmixer`.

The hyperparameters for each model on different datasets follow the official configurations provided in their corresponding GitHub repositories. For the PatchTST model on the Solar-Energy dataset, since no official configuration was provided, we adopted the hyperparameter settings from iTransformer.

# C    Model Detail

## C.1    ESP

---

**Algorithm 1:** Embedding-Similarity Penalty (ESP) for Time Series Forecasting

---

**Input:** Mini–batch $\mathcal{B} = \{(X_i, Y_i)\}_{i=1}^n$, encoder $f_{\mathrm{enc}}$, predictor $f_{\mathrm{pred}}$
**Output:** Penalty loss $\mathcal{L}_{\mathrm{ESP}}$

**1  1. Forward pass to compute encoder outputs**
**2  for** $i \leftarrow 1$ **to** $n$ **do**
**3**      $Z_i \leftarrow f_{\mathrm{enc}}(X_i)$                                          #encoder output $\in \mathbb{R}^{L \times D}$
**4  end**
**5  2. Compute pairwise Frobenius distances**
**6  Initialize** $\Delta^E, \Delta^O \in \mathbb{R}^{n \times n}$
**7  for** $i \leftarrow 1$ **to** $n$ **do**
**8**      **for** $j \leftarrow i$ **to** $n$ **do**
**9**          $\Delta_{ij}^E \leftarrow \frac{1}{L \times D} \|Z_i - Z_j\|_F^2$                          #embedding similarity
**10**         $\Delta_{ij}^O \leftarrow \frac{1}{P \times D} \|Y_i - Y_j\|_F^2$                          #output similarity
**11**         $\Delta_{ji}^E \leftarrow \Delta_{ij}^E,$
**12**         $\Delta_{ji}^O \leftarrow \Delta_{ij}^O$                                          #symmetry
**13**     **end**
**14  end**
**15  3. Compute pairwise penalties**
**16  for** $i \leftarrow 1$ **to** $n$ **do**
**17**     **for** $j \leftarrow 1$ **to** $n$ **do**
**18**         $P_{ij} \leftarrow |\Delta_{ij}^E - \Delta_{ij}^O|_+$                          #element-wise consistency penalty
**19**     **end**
**20  end**
**21  4. Compute final regularization loss**
**22**  $\mathcal{L}_{\mathrm{ESP}} \leftarrow \frac{1}{n^2} \sum_{i=1}^n \sum_{j=1}^n P_{ij}$
**23  5. Backward pass and update**
**24  Update** $\theta$ using forecasting loss $+\lambda_{\mathrm{ESP}} \cdot \mathcal{L}_{\mathrm{ESP}}$

---

# D  Full Results

## D.1  Experimental Result Details

Table 5: Multivariate forecasting results with prediction lengths $H \in \{48, 72, 96, 120, 144, 168, 192\}$ and fixed input window length $L = 48$. Red highlights indicate performance improvements $> 0.005$ using our method, while blue highlights denote improvements $> 0$ but $\leq 0.005$.

| Models | | SOFTS | | | | TimeMixer | | | | iTransformer | | | | PatchTST | | | | TSMixer | | | |
|---|---|---|---|---|---|---|---|---|---|---|---|---|---|---|---|---|---|---|---|---|---|
| | | original | | AMRC | | original | | AMRC | | original | | AMRC | | original | | AMRC | | original | | AMRC | |
| Metric | | MSE | MAE | MSE | MAE | MSE | MAE | MSE | MAE | MSE | MAE | MSE | MAE | MSE | MAE | MSE | MAE | MSE | MAE | MSE | MAE |
| ETTh1 | 48 | 0.354 | 0.381 | 0.334 | 0.359 | 0.333 | 0.372 | 0.324 | 0.365 | 0.353 | 0.381 | 0.344 | 0.365 | 0.373 | 0.394 | 0.363 | 0.388 | 0.345 | 0.375 | 0.331 | 0.361 |
| | 72 | 0.379 | 0.397 | 0.364 | 0.380 | 0.361 | 0.389 | 0.356 | 0.384 | 0.381 | 0.396 | 0.367 | 0.377 | 0.387 | 0.406 | 0.375 | 0.396 | 0.376 | 0.395 | 0.363 | 0.382 |
| | 96 | 0.394 | 0.407 | 0.377 | 0.388 | 0.376 | 0.399 | 0.372 | 0.394 | 0.401 | 0.408 | 0.393 | 0.387 | 0.411 | 0.417 | 0.397 | 0.405 | 0.389 | 0.405 | 0.376 | 0.393 |
| | 120 | 0.418 | 0.421 | 0.400 | 0.401 | 0.398 | 0.410 | 0.397 | 0.404 | 0.419 | 0.419 | 0.410 | 0.403 | 0.428 | 0.426 | 0.415 | 0.418 | 0.406 | 0.415 | 0.386 | 0.401 |
| | 144 | 0.426 | 0.425 | 0.404 | 0.402 | 0.416 | 0.421 | 0.412 | 0.413 | 0.434 | 0.427 | 0.420 | 0.409 | 0.443 | 0.434 | 0.432 | 0.424 | 0.419 | 0.422 | 0.406 | 0.405 |
| | 168 | 0.438 | 0.434 | 0.416 | 0.409 | 0.426 | 0.427 | 0.420 | 0.417 | 0.443 | 0.434 | 0.431 | 0.421 | 0.456 | 0.441 | 0.441 | 0.429 | 0.433 | 0.431 | 0.412 | 0.416 |
| | 192 | 0.450 | 0.435 | 0.427 | 0.410 | 0.439 | 0.435 | 0.435 | 0.430 | 0.458 | 0.443 | 0.449 | 0.431 | 0.468 | 0.448 | 0.455 | 0.444 | 0.446 | 0.440 | 0.427 | 0.421 |
| | Avg | 0.408 | 0.414 | 0.389 | 0.393 | 0.393 | 0.408 | 0.388 | 0.401 | 0.413 | 0.415 | 0.402 | 0.399 | 0.424 | 0.424 | 0.411 | 0.415 | 0.402 | 0.412 | 0.386 | 0.397 |
| ETTh2 | 48 | 0.236 | 0.304 | 0.221 | 0.303 | 0.235 | 0.302 | 0.230 | 0.290 | 0.246 | 0.312 | 0.237 | 0.306 | 0.241 | 0.306 | 0.229 | 0.301 | 0.241 | 0.302 | 0.229 | 0.277 |
| | 72 | 0.281 | 0.333 | 0.275 | 0.344 | 0.273 | 0.326 | 0.269 | 0.309 | 0.283 | 0.336 | 0.280 | 0.327 | 0.281 | 0.332 | 0.274 | 0.328 | 0.276 | 0.328 | 0.270 | 0.299 |
| | 96 | 0.319 | 0.356 | 0.307 | 0.364 | 0.298 | 0.343 | 0.294 | 0.328 | 0.309 | 0.352 | 0.306 | 0.345 | 0.307 | 0.349 | 0.299 | 0.349 | 0.303 | 0.345 | 0.298 | 0.314 |
| | 120 | 0.328 | 0.361 | 0.315 | 0.368 | 0.323 | 0.359 | 0.321 | 0.342 | 0.332 | 0.366 | 0.329 | 0.359 | 0.331 | 0.361 | 0.326 | 0.357 | 0.331 | 0.361 | 0.324 | 0.332 |
| | 144 | 0.354 | 0.375 | 0.333 | 0.371 | 0.343 | 0.371 | 0.343 | 0.352 | 0.354 | 0.377 | 0.346 | 0.368 | 0.353 | 0.374 | 0.347 | 0.372 | 0.353 | 0.373 | 0.352 | 0.338 |
| | 168 | 0.371 | 0.386 | 0.354 | 0.385 | 0.368 | 0.388 | 0.369 | 0.370 | 0.379 | 0.391 | 0.376 | 0.392 | 0.376 | 0.387 | 0.367 | 0.390 | 0.370 | 0.485 | 0.369 | 0.449 |
| | 192 | 0.391 | 0.399 | 0.373 | 0.399 | 0.386 | 0.399 | 0.387 | 0.381 | 0.398 | 0.402 | 0.393 | 0.395 | 0.397 | 0.399 | 0.391 | 0.396 | 0.396 | 0.402 | 0.390 | 0.369 |
| | Avg | 0.326 | 0.359 | 0.311 | 0.362 | 0.318 | 0.355 | 0.316 | 0.339 | 0.329 | 0.362 | 0.324 | 0.356 | 0.327 | 0.358 | 0.319 | 0.356 | 0.324 | 0.357 | 0.319 | 0.340 |
| ETTm1 | 48 | 0.497 | 0.434 | 0.487 | 0.422 | 0.462 | 0.423 | 0.443 | 0.397 | 0.543 | 0.453 | 0.529 | 0.448 | 0.481 | 0.424 | 0.472 | 0.417 | 0.452 | 0.411 | 0.442 | 0.404 |
| | 72 | 0.462 | 0.421 | 0.457 | 0.414 | 0.453 | 0.420 | 0.438 | 0.394 | 0.497 | 0.438 | 0.479 | 0.438 | 0.443 | 0.411 | 0.438 | 0.400 | 0.419 | 0.399 | 0.406 | 0.399 |
| | 96 | 0.447 | 0.418 | 0.440 | 0.409 | 0.437 | 0.415 | 0.418 | 0.392 | 0.475 | 0.431 | 0.461 | 0.429 | 0.422 | 0.402 | 0.417 | 0.389 | 0.404 | 0.395 | 0.398 | 0.394 |
| | 120 | 0.478 | 0.432 | 0.470 | 0.422 | 0.473 | 0.432 | 0.452 | 0.407 | 0.512 | 0.447 | 0.499 | 0.449 | 0.459 | 0.420 | 0.455 | 0.413 | 0.438 | 0.413 | 0.431 | 0.412 |
| | 144 | 0.507 | 0.448 | 0.495 | 0.434 | 0.489 | 0.441 | 0.469 | 0.419 | 0.542 | 0.461 | 0.525 | 0.455 | 0.481 | 0.434 | 0.476 | 0.426 | 0.460 | 0.425 | 0.451 | 0.424 |
| | 168 | 0.498 | 0.443 | 0.486 | 0.429 | 0.479 | 0.437 | 0.461 | 0.416 | 0.531 | 0.457 | 0.517 | 0.458 | 0.477 | 0.433 | 0.474 | 0.429 | 0.457 | 0.426 | 0.452 | 0.430 |
| | 192 | 0.501 | 0.445 | 0.488 | 0.430 | 0.470 | 0.435 | 0.447 | 0.409 | 0.521 | 0.452 | 0.504 | 0.453 | 0.465 | 0.427 | 0.459 | 0.418 | 0.451 | 0.422 | 0.444 | 0.421 |
| | Avg | 0.484 | 0.434 | 0.475 | 0.423 | 0.466 | 0.429 | 0.447 | 0.405 | 0.517 | 0.448 | 0.502 | 0.447 | 0.461 | 0.422 | 0.456 | 0.413 | 0.440 | 0.413 | 0.432 | 0.412 |
| ETTm2 | 48 | 0.154 | 0.246 | 0.141 | 0.226 | 0.157 | 0.251 | 0.150 | 0.235 | 0.159 | 0.255 | 0.158 | 0.242 | 0.160 | 0.253 | 0.151 | 0.241 | 0.147 | 0.238 | 0.141 | 0.218 |
| | 72 | 0.174 | 0.261 | 0.166 | 0.246 | 0.173 | 0.261 | 0.164 | 0.249 | 0.178 | 0.268 | 0.178 | 0.261 | 0.176 | 0.265 | 0.159 | 0.253 | 0.170 | 0.257 | 0.158 | 0.233 |
| | 96 | 0.189 | 0.271 | 0.179 | 0.254 | 0.190 | 0.274 | 0.186 | 0.256 | 0.193 | 0.276 | 0.188 | 0.268 | 0.190 | 0.272 | 0.176 | 0.253 | 0.186 | 0.265 | 0.177 | 0.247 |
| | 120 | 0.211 | 0.287 | 0.200 | 0.269 | 0.210 | 0.285 | 0.209 | 0.268 | 0.214 | 0.290 | 0.210 | 0.280 | 0.212 | 0.287 | 0.194 | 0.273 | 0.208 | 0.282 | 0.198 | 0.261 |
| | 144 | 0.236 | 0.302 | 0.221 | 0.280 | 0.231 | 0.300 | 0.226 | 0.283 | 0.236 | 0.304 | 0.235 | 0.296 | 0.233 | 0.302 | 0.220 | 0.286 | 0.228 | 0.296 | 0.219 | 0.272 |
| | 168 | 0.248 | 0.311 | 0.233 | 0.289 | 0.245 | 0.311 | 0.242 | 0.295 | 0.251 | 0.313 | 0.251 | 0.301 | 0.248 | 0.310 | 0.232 | 0.291 | 0.244 | 0.305 | 0.235 | 0.279 |
| | 192 | 0.261 | 0.316 | 0.245 | 0.293 | 0.255 | 0.313 | 0.250 | 0.295 | 0.263 | 0.321 | 0.257 | 0.312 | 0.260 | 0.317 | 0.240 | 0.300 | 0.257 | 0.313 | 0.243 | 0.290 |
| | Avg | 0.210 | 0.285 | 0.198 | 0.265 | 0.209 | 0.285 | 0.204 | 0.269 | 0.213 | 0.290 | 0.211 | 0.280 | 0.211 | 0.287 | 0.196 | 0.271 | 0.201 | 0.279 | 0.196 | 0.257 |
| Solar-Energy | 48 | 0.256 | 0.294 | 0.253 | 0.289 | 0.264 | 0.296 | 0.259 | 0.292 | 0.357 | 0.344 | 0.354 | 0.337 | 0.362 | 0.386 | 0.347 | 0.378 | 0.248 | 0.283 | 0.240 | 0.282 |
| | 72 | 0.311 | 0.333 | 0.313 | 0.333 | 0.293 | 0.341 | 0.292 | 0.342 | 0.441 | 0.381 | 0.442 | 0.373 | 0.429 | 0.430 | 0.418 | 0.425 | 0.305 | 0.327 | 0.298 | 0.328 |
| | 96 | 0.308 | 0.324 | 0.308 | 0.322 | 0.309 | 0.343 | 0.304 | 0.342 | 0.446 | 0.374 | 0.443 | 0.363 | 0.409 | 0.417 | 0.392 | 0.409 | 0.308 | 0.334 | 0.301 | 0.346 |
| | 120 | 0.283 | 0.302 | 0.282 | 0.299 | 0.288 | 0.307 | 0.283 | 0.311 | 0.385 | 0.345 | 0.382 | 0.330 | 0.364 | 0.376 | 0.353 | 0.369 | 0.290 | 0.315 | 0.283 | 0.309 |
| | 144 | 0.296 | 0.316 | 0.291 | 0.309 | 0.288 | 0.305 | 0.284 | 0.305 | 0.369 | 0.331 | 0.366 | 0.322 | 0.344 | 0.355 | 0.331 | 0.351 | 0.280 | 0.304 | 0.275 | 0.301 |
| | 168 | 0.293 | 0.311 | 0.288 | 0.304 | 0.279 | 0.307 | 0.273 | 0.309 | 0.373 | 0.337 | 0.367 | 0.326 | 0.339 | 0.356 | 0.326 | 0.347 | 0.286 | 0.312 | 0.274 | 0.306 |
| | 192 | 0.304 | 0.316 | 0.298 | 0.308 | 0.296 | 0.317 | 0.293 | 0.319 | 0.392 | 0.351 | 0.391 | 0.342 | 0.369 | 0.356 | 0.360 | 0.352 | 0.297 | 0.321 | 0.289 | 0.319 |
| | Avg | 0.293 | 0.314 | 0.290 | 0.309 | 0.288 | 0.317 | 0.284 | 0.317 | 0.395 | 0.352 | 0.392 | 0.342 | 0.374 | 0.383 | 0.361 | 0.376 | 0.288 | 0.314 | 0.280 | 0.313 |
| Weather | 48 | 0.161 | 0.188 | 0.152 | 0.174 | 0.153 | 0.189 | 0.143 | 0.182 | 0.159 | 0.189 | 0.147 | 0.177 | 0.189 | 0.264 | 0.183 | 0.245 | 0.169 | 0.237 | 0.158 | 0.226 |
| | 72 | 0.178 | 0.212 | 0.174 | 0.212 | 0.179 | 0.219 | 0.165 | 0.208 | 0.189 | 0.211 | 0.189 | 0.204 | 0.208 | 0.279 | 0.202 | 0.262 | 0.200 | 0.273 | 0.196 | 0.265 |
| | 96 | 0.201 | 0.232 | 0.195 | 0.221 | 0.203 | 0.251 | 0.191 | 0.243 | 0.201 | 0.234 | 0.197 | 0.225 | 0.219 | 0.288 | 0.214 | 0.276 | 0.223 | 0.298 | 0.215 | 0.289 |
| | 120 | 0.204 | 0.235 | 0.197 | 0.223 | 0.195 | 0.237 | 0.185 | 0.227 | 0.213 | 0.202 | 0.205 | 0.196 | 0.222 | 0.291 | 0.217 | 0.274 | 0.228 | 0.300 | 0.214 | 0.299 |
| | 144 | 0.221 | 0.249 | 0.210 | 0.233 | 0.202 | 0.243 | 0.193 | 0.232 | 0.219 | 0.247 | 0.212 | 0.238 | 0.215 | 0.287 | 0.215 | 0.273 | 0.236 | 0.313 | 0.224 | 0.304 |
| | 168 | 0.224 | 0.254 | 0.213 | 0.238 | 0.212 | 0.251 | 0.206 | 0.243 | 0.233 | 0.258 | 0.229 | 0.247 | 0.237 | 0.263 | 0.234 | 0.248 | 0.235 | 0.261 | 0.241 | 0.255 |
| | 192 | 0.244 | 0.266 | 0.232 | 0.249 | 0.233 | 0.269 | 0.219 | 0.261 | 0.245 | 0.271 | 0.241 | 0.260 | 0.214 | 0.288 | 0.205 | 0.270 | 0.263 | 0.331 | 0.235 | 0.329 |
| | Avg | 0.205 | 0.234 | 0.196 | 0.220 | 0.197 | 0.237 | 0.186 | 0.228 | 0.209 | 0.237 | 0.201 | 0.221 | 0.215 | 0.280 | 0.210 | 0.264 | 0.222 | 0.288 | 0.212 | 0.281 |
| ECL | 48 | 0.146 | 0.233 | 0.138 | 0.221 | 0.172 | 0.259 | 0.164 | 0.256 | 0.151 | 0.238 | 0.136 | 0.216 | 0.189 | 0.264 | 0.188 | 0.264 | 0.148 | 0.236 | 0.141 | 0.224 |
| | 72 | 0.161 | 0.247 | 0.158 | 0.241 | 0.188 | 0.274 | 0.183 | 0.272 | 0.168 | 0.253 | 0.158 | 0.228 | 0.208 | 0.279 | 0.202 | 0.278 | 0.165 | 0.251 | 0.157 | 0.236 |
| | 96 | 0.171 | 0.256 | 0.166 | 0.248 | 0.199 | 0.284 | 0.194 | 0.283 | 0.178 | 0.262 | 0.161 | 0.236 | 0.219 | 0.288 | 0.211 | 0.294 | 0.175 | 0.260 | 0.172 | 0.248 |
| | 120 | 0.176 | 0.261 | 0.170 | 0.251 | 0.203 | 0.287 | 0.194 | 0.284 | 0.183 | 0.267 | 0.172 | 0.249 | 0.222 | 0.291 | 0.221 | 0.290 | 0.180 | 0.265 | 0.177 | 0.257 |
| | 144 | 0.175 | 0.261 | 0.165 | 0.247 | 0.200 | 0.283 | 0.191 | 0.281 | 0.182 | 0.267 | 0.173 | 0.245 | 0.215 | 0.287 | 0.210 | 0.291 | 0.180 | 0.265 | 0.176 | 0.254 |
| | 168 | 0.176 | 0.262 | 0.166 | 0.248 | 0.199 | 0.285 | 0.194 | 0.284 | 0.182 | 0.266 | 0.165 | 0.249 | 0.211 | 0.284 | 0.206 | 0.286 | 0.181 | 0.265 | 0.177 | 0.252 |
| | 192 | 0.181 | 0.266 | 0.170 | 0.252 | 0.200 | 0.283 | 0.196 | 0.279 | 0.186 | 0.270 | 0.176 | 0.250 | 0.214 | 0.288 | 0.212 | 0.291 | 0.184 | 0.267 | 0.184 | 0.258 |
| | Avg | 0.169 | 0.255 | 0.162 | 0.244 | 0.194 | 0.279 | 0.188 | 0.277 | 0.176 | 0.260 | 0.163 | 0.239 | 0.211 | 0.283 | 0.207 | 0.285 | 0.173 | 0.258 | 0.169 | 0.247 |

Table 6: Detailed statistical analysis of AMRC effectiveness. This table presents the mean $\pm$ standard deviation over 10 runs for original and AMRC-enhanced models. The 'Conf (%)' row indicates the confidence level from significance tests comparing AMRC to the baseline.

| Model | Metric | ETTh1 | | ETTh2 | | ETTm1 | | ETTm2 | |
|---|---|---|---|---|---|---|---|---|---|
| | | MSE | MAE | MSE | MAE | MSE | MAE | MSE | MAE |
| SOFTS | Original | $0.408 \pm 0.004$ | $0.414 \pm 0.003$ | $0.326 \pm 0.003$ | $0.359 \pm 0.004$ | $0.484 \pm 0.004$ | $0.434 \pm 0.003$ | $0.210 \pm 0.002$ | $0.285 \pm 0.004$ |
| | AMRC | $0.389 \pm 0.011$ | $0.393 \pm 0.009$ | $0.311 \pm 0.008$ | $0.362 \pm 0.004$ | $0.475 \pm 0.006$ | $0.423 \pm 0.005$ | $0.198 \pm 0.007$ | $0.265 \pm 0.006$ |
| | Conf (%) | 99 | 99 | 99 | 95 | 99 | 99 | 99 | 99 |
| iTransformer | Original | $0.413 \pm 0.001$ | $0.415 \pm 0.002$ | $0.329 \pm 0.002$ | $0.362 \pm 0.002$ | $0.517 \pm 0.003$ | $0.448 \pm 0.001$ | $0.213 \pm 0.001$ | $0.290 \pm 0.002$ |
| | AMRC | $0.402 \pm 0.004$ | $0.399 \pm 0.005$ | $0.324 \pm 0.005$ | $0.356 \pm 0.004$ | $0.502 \pm 0.004$ | $0.447 \pm 0.002$ | $0.211 \pm 0.003$ | $0.280 \pm 0.003$ |
| | Conf (%) | 99 | 99 | 95 | 95 | 99 | 95 | 95 | 99 |
| TimeMixer | Original | $0.393 \pm 0.003$ | $0.408 \pm 0.005$ | $0.318 \pm 0.006$ | $0.355 \pm 0.008$ | $0.466 \pm 0.004$ | $0.429 \pm 0.006$ | $0.209 \pm 0.002$ | $0.285 \pm 0.004$ |
| | AMRC | $0.388 \pm 0.006$ | $0.401 \pm 0.007$ | $0.316 \pm 0.008$ | $0.339 \pm 0.007$ | $0.447 \pm 0.008$ | $0.405 \pm 0.009$ | $0.204 \pm 0.007$ | $0.269 \pm 0.006$ |
| | Conf (%) | 99 | 99 | 99 | 99 | 99 | 99 | 99 | 99 |
| PatchTST | Original | $0.424 \pm 0.003$ | $0.424 \pm 0.002$ | $0.327 \pm 0.001$ | $0.358 \pm 0.003$ | $0.461 \pm 0.003$ | $0.422 \pm 0.002$ | $0.211 \pm 0.002$ | $0.287 \pm 0.003$ |
| | AMRC | $0.411 \pm 0.005$ | $0.415 \pm 0.003$ | $0.319 \pm 0.004$ | $0.356 \pm 0.004$ | $0.456 \pm 0.004$ | $0.413 \pm 0.003$ | $0.196 \pm 0.005$ | $0.271 \pm 0.004$ |
| | Conf (%) | 99 | 99 | 99 | 95 | 99 | 99 | 99 | 99 |
| TSMixer | Original | $0.402 \pm 0.003$ | $0.412 \pm 0.005$ | $0.324 \pm 0.004$ | $0.357 \pm 0.004$ | $0.440 \pm 0.003$ | $0.413 \pm 0.006$ | $0.201 \pm 0.005$ | $0.279 \pm 0.003$ |
| | AMRC | $0.386 \pm 0.010$ | $0.397 \pm 0.008$ | $0.319 \pm 0.007$ | $0.340 \pm 0.011$ | $0.432 \pm 0.010$ | $0.412 \pm 0.006$ | $0.196 \pm 0.007$ | $0.257 \pm 0.013$ |
| | Conf (%) | 99 | 99 | 99 | 99 | 99 | 95 | 95 | 99 |

| Model | Metric | Solar-Energy | | Electricity | | Weather | |
|---|---|---|---|---|---|---|---|
| | | MSE | MAE | MSE | MAE | MSE | MAE |
| SOFTS | Original | $0.293 \pm 0.003$ | $0.314 \pm 0.004$ | $0.169 \pm 0.003$ | $0.255 \pm 0.004$ | $0.205 \pm 0.002$ | $0.234 \pm 0.003$ |
| | AMRC | $0.290 \pm 0.007$ | $0.309 \pm 0.007$ | $0.162 \pm 0.006$ | $0.244 \pm 0.007$ | $0.196 \pm 0.005$ | $0.186 \pm 0.004$ |
| | Conf (%) | 95 | 95 | 99 | 99 | 99 | 99 |
| iTransformer | Original | $0.395 \pm 0.002$ | $0.352 \pm 0.002$ | $0.176 \pm 0.002$ | $0.260 \pm 0.003$ | $0.209 \pm 0.003$ | $0.237 \pm 0.002$ |
| | AMRC | $0.392 \pm 0.006$ | $0.342 \pm 0.005$ | $0.163 \pm 0.004$ | $0.239 \pm 0.007$ | $0.201 \pm 0.005$ | $0.221 \pm 0.008$ |
| | Conf (%) | 95 | 99 | 99 | 99 | 99 | 99 |
| TimeMixer | Original | $0.288 \pm 0.003$ | $0.317 \pm 0.000$ | $0.194 \pm 0.010$ | $0.279 \pm 0.006$ | $0.197 \pm 0.010$ | $0.237 \pm 0.009$ |
| | AMRC | $0.284 \pm 0.008$ | $0.317 \pm 0.008$ | $0.188 \pm 0.012$ | $0.277 \pm 0.008$ | $0.186 \pm 0.014$ | $0.228 \pm 0.011$ |
| | Conf (%) | 95 | 90 | 99 | 95 | 99 | 99 |
| PatchTST | Original | $0.374 \pm 0.003$ | $0.383 \pm 0.004$ | $0.211 \pm 0.002$ | $0.283 \pm 0.002$ | $0.215 \pm 0.002$ | $0.280 \pm 0.003$ |
| | AMRC | $0.361 \pm 0.006$ | $0.376 \pm 0.007$ | $0.207 \pm 0.004$ | $0.285 \pm 0.002$ | $0.210 \pm 0.003$ | $0.264 \pm 0.003$ |
| | Conf (%) | 95 | 99 | 99 | 95 | 99 | 99 |
| TSMixer | Original | $0.288 \pm 0.004$ | $0.314 \pm 0.004$ | $0.173 \pm 0.005$ | $0.258 \pm 0.006$ | $0.222 \pm 0.002$ | $0.288 \pm 0.007$ |
| | AMRC | $0.280 \pm 0.011$ | $0.313 \pm 0.005$ | $0.169 \pm 0.009$ | $0.247 \pm 0.006$ | $0.212 \pm 0.010$ | $0.281 \pm 0.009$ |
| | Conf (%) | 99 | 95 | 99 | 95 | 99 | 99 |

Table 7: Additional experiments on the Illness and ExchangeRate datasets using the SOFTS backbone. Results are reported as mean $\pm$ standard deviation over 10 runs. Due to the small size of the Illness dataset (967 samples), the experimental setup was adjusted (Input $L = 48$, prediction lengths $H \in \{24, 36, 48, 60\}$) following the PatchTST protocol [17].

| | $H$ | AMRC MSE | AMRC MAE | Original MSE | Original MAE | Conf-MSE (%) | Conf-MAE (%) |
|---|---|---|---|---|---|---|---|
| Illness | 24 | $1.633 \pm 0.07$ | $0.789 \pm 0.06$ | $1.776 \pm 0.14$ | $0.852 \pm 0.03$ | 95 | 95 |
| | 36 | $1.858 \pm 0.07$ | $0.854 \pm 0.06$ | $1.942 \pm 0.12$ | $0.904 \pm 0.04$ | 90 | 95 |
| | 48 | $2.035 \pm 0.07$ | $0.916 \pm 0.05$ | $2.153 \pm 0.12$ | $0.954 \pm 0.03$ | 95 | 95 |
| | 60 | $2.054 \pm 0.07$ | $0.935 \pm 0.03$ | $2.113 \pm 0.10$ | $0.958 \pm 0.03$ | 90 | 90 |
| ExchangeRate | 48 | $0.03913 \pm 0.001$ | $0.13016 \pm 0.007$ | $0.04208 \pm 0.001$ | $0.13728 \pm 0.008$ | 99 | 95 |
| | 72 | $0.05788 \pm 0.002$ | $0.16432 \pm 0.008$ | $0.06093 \pm 0.003$ | $0.17116 \pm 0.009$ | 95 | 90 |
| | 96 | $0.07927 \pm 0.002$ | $0.18782 \pm 0.005$ | $0.08329 \pm 0.004$ | $0.20196 \pm 0.001$ | 99 | 99 |
| | 120 | $0.10053 \pm 0.001$ | $0.21698 \pm 0.002$ | $0.10695 \pm 0.001$ | $0.22840 \pm 0.001$ | 99 | 99 |
| | 144 | $0.12417 \pm 0.002$ | $0.24484 \pm 0.002$ | $0.12935 \pm 0.002$ | $0.25241 \pm 0.001$ | 99 | 99 |
| | 168 | $0.14602 \pm 0.002$ | $0.25976 \pm 0.001$ | $0.16047 \pm 0.003$ | $0.28234 \pm 0.003$ | 99 | 99 |
| | 192 | $0.17457 \pm 0.007$ | $0.29181 \pm 0.006$ | $0.18376 \pm 0.007$ | $0.30397 \pm 0.007$ | 95 | 99 |

Table 8: Hyperparameter sensitivity analysis for the mask count ($m$) on the ETTh1 dataset with the iTransformer backbone. We report the average MSE and MAE as $m$ varies. The results show diminishing returns, justifying our choice of $m = 12$.

| Model | $m$/L | MSE (mean) | MAE (mean) |
|---|---|---|---|
| SOFTS | 1/8 | 0.407 | 0.416 |
| | 1/6 | 0.407 | 0.409 |
| | 1/4 | 0.389 | 0.393 |
| | 1/3 | 0.384 | 0.388 |
| | 1/2 | 0.381 | 0.385 |
| | 3/4 | 0.380 | 0.384 |
| iTransformer | 1/8 | 0.412 | 0.412 |
| | 1/6 | 0.411 | 0.417 |
| | 1/4 | 0.402 | 0.399 |
| | 1/3 | 0.398 | 0.395 |
| | 1/2 | 0.395 | 0.392 |
| | 3/4 | 0.394 | 0.390 |
| TimeMixer | 1/8 | 0.396 | 0.405 |
| | 1/6 | 0.391 | 0.409 |
| | 1/4 | 0.388 | 0.401 |
| | 1/3 | 0.385 | 0.398 |
| | 1/2 | 0.383 | 0.396 |
| | 3/4 | 0.382 | 0.395 |
| PatchTST | 1/8 | 0.422 | 0.420 |
| | 1/6 | 0.419 | 0.421 |
| | 1/4 | 0.411 | 0.415 |
| | 1/3 | 0.406 | 0.411 |
| | 1/2 | 0.403 | 0.409 |
| | 3/4 | 0.401 | 0.408 |
| TSMixer | 1/8 | 0.401 | 0.411 |
| | 1/6 | 0.395 | 0.408 |
| | 1/4 | 0.386 | 0.397 |
| | 1/3 | 0.381 | 0.392 |
| | 1/2 | 0.378 | 0.389 |
| | 3/4 | 0.376 | 0.387 |

Table 9: The robustness of AMRC on SOFTS. Results are averaged over ten experiments, each tested with different random seeds.

| Dataset | ETTh1 | | ETTh2 | | Solar-Energy | | Weather | |
|---|---|---|---|---|---|---|---|---|
| Prediction | MSE | MAE | MSE | MAE | MSE | MAE | MSE | MAE |
| 48 | $0.334 \pm 0.003$ | $0.359 \pm 0.002$ | $0.221 \pm 0.001$ | $0.303 \pm 0.002$ | $0.253 \pm 0.002$ | $0.289 \pm 0.002$ | $0.152 \pm 0.001$ | $0.174 \pm 0.005$ |
| 72 | $0.364 \pm 0.001$ | $0.380 \pm 0.001$ | $0.275 \pm 0.002$ | $0.344 \pm 0.001$ | $0.313 \pm 0.001$ | $0.333 \pm 0.001$ | $0.174 \pm 0.003$ | $0.203 \pm 0.002$ |
| 96 | $0.377 \pm 0.002$ | $0.388 \pm 0.002$ | $0.307 \pm 0.002$ | $0.364 \pm 0.001$ | $0.308 \pm 0.002$ | $0.322 \pm 0.002$ | $0.195 \pm 0.002$ | $0.221 \pm 0.002$ |
| 120 | $0.400 \pm 0.002$ | $0.400 \pm 0.005$ | $0.315 \pm 0.001$ | $0.368 \pm 0.002$ | $0.282 \pm 0.002$ | $0.299 \pm 0.002$ | $0.197 \pm 0.001$ | $0.223 \pm 0.003$ |
| 144 | $0.404 \pm 0.002$ | $0.402 \pm 0.002$ | $0.333 \pm 0.002$ | $0.371 \pm 0.002$ | $0.291 \pm 0.002$ | $0.309 \pm 0.003$ | $0.210 \pm 0.002$ | $0.233 \pm 0.001$ |
| 168 | $0.416 \pm 0.002$ | $0.409 \pm 0.002$ | $0.354 \pm 0.002$ | $0.385 \pm 0.001$ | $0.288 \pm 0.002$ | $0.304 \pm 0.002$ | $0.213 \pm 0.003$ | $0.238 \pm 0.002$ |
| 192 | $0.427 \pm 0.002$ | $0.410 \pm 0.002$ | $0.373 \pm 0.002$ | $0.399 \pm 0.005$ | $0.298 \pm 0.001$ | $0.308 \pm 0.001$ | $0.232 \pm 0.003$ | $0.249 \pm 0.002$ |

Table 10: AMRC Effectiveness with Ideal Masking Averaged Across All Input Lengths. Ratio is the percentage of samples with reduced MSE under ideal masking. Ratio* is the same metric after training with AMRC. These results are averaged across all input lengths ($L \in \{24, 48, 96, 120, 144, 168, 192\}$) to show overall robustness.

| Models | SOFTS | | TimeMixer | | iTransformer | | PatchTST | | TSMixer | |
|---|---|---|---|---|---|---|---|---|---|---|
| Metric | Ratio | Ratio* | Ratio | Ratio* | Ratio | Ratio* | Ratio | Ratio* | Ratio | Ratio* |
| ETTh1 | 57.14% | 48.7% | 49.69% | 37.81% | 51.88% | 42.93% | 57.81% | 42.91% | 52.92% | 39.1% |
| ETTh2 | 30.99% | 21.09% | 47.16% | 34.89% | 33.28% | 24.11% | 43.54% | 27.91% | 44.29% | 29.63% |
| Solar-Energy | 44.87% | 33.83% | 37.52% | 29.61% | 71.18% | 67.72% | 53.26% | 48.02% | 41.66% | 30.11% |
| Weather | 54.63% | 48.33% | 67.39% | 52.78% | 79.4% | 69.36% | 41.98% | 29.86% | 69.32% | 56.26% |

## D.2 Visualized Prediction Comparison Chart

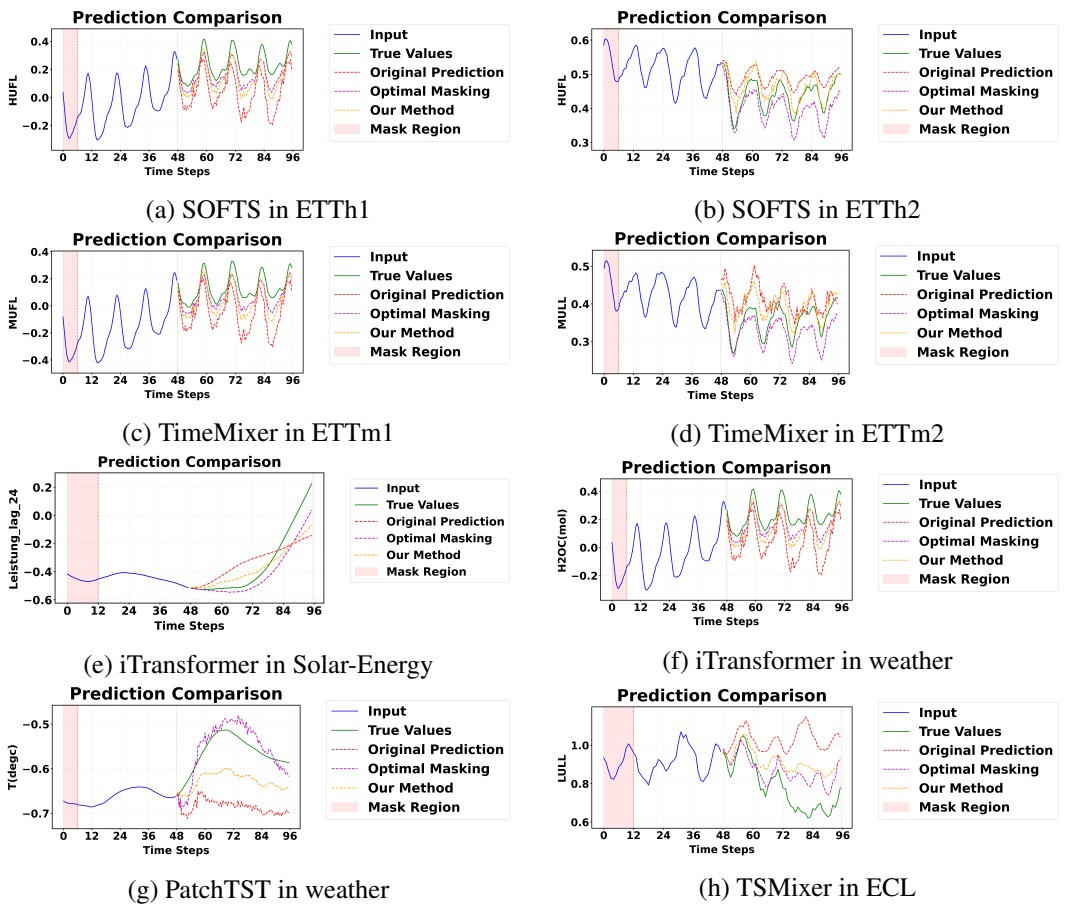

(a) SOFTS in ETTh1

(b) SOFTS in ETTh2

(c) TimeMixer in ETTm1

(d) TimeMixer in ETTm2

(e) iTransformer in Solar-Energy

(f) iTransformer in weather

(g) PatchTST in weather

(h) TSMixer in ECL

Figure 4: Qualitative comparison of prediction performance. Each subplot provides a visual comparison of the ground truth, the baseline model, the optimal masking result, and the forecast from AMRC on a specific model and dataset. The mask region highlights the prefix portion of the input.

# E Dataset description

Here we provide detailed descriptions along with download links for each dataset:

1. **ETT (Electricity Transformer Temperature)** [37][4]: This collection includes two hourly-resolution datasets (ETTh) and two 15-minute-resolution datasets (ETTm). Each dataset captures seven key operational metrics (including oil and load measurements) from electricity transformers, spanning from July 2016 to July 2018.

2. **Electricity**[5]: Comprising hourly power consumption records from 321 customers, this dataset covers the period from 2012 to 2014.

3. **Weather**: Featuring 21 meteorological indicators (such as air temperature and humidity), this dataset provides 10-minute-interval recordings throughout 2020, sourced from weather stations in Germany.

4. **Solar-Energy**: Documents the solar power generation output of 137 photovoltaic plants in 2006, with measurements taken at 10-minute intervals.

Table 11: Detailed Dataset Descriptions. The table summarizes key characteristics of the time series datasets, including the number of channels, prediction lengths, dataset splits, temporal granularity, and application domains.

| Dataset | Channels | Prediction Length | Dataset Split (Train, Val, Test) | Granularity | Domain |
|---------|----------|-------------------|----------------------------------|-------------|--------|
| ETTh1, ETTh2 | 7 | {48, 72, 96, 120, 144, 168, 192} | (8545, 2881, 2881) | Hourly | Electricity |
| ETTm1, ETTm2 | 7 | {48, 72, 96, 120, 144, 168, 192} | (34465, 11521, 11521) | 15min | Electricity |
| Weather | 21 | {48, 72, 96, 120, 144, 168, 192} | (36792, 5271, 10540) | 10min | Weather |
| ECL | 321 | {48, 72, 96, 120, 144, 168, 192} | (18317, 2633, 5261) | Hourly | Electricity |
| Solar-Energy | 137 | {48, 72, 96, 120, 144, 168, 192} | (36601, 5161, 10417) | 10min | Energy |

---

[4]https://github.com/zhouhaoyi/ETDataset
[5]https://archive.ics.uci.edu/dataset/321/electricityloaddiagrams20112014

