# OpenReview forum: "Abstain Mask Retain Core: Time Series Prediction by Adaptive Masking Loss with Representation Consistency"
_NeurIPS.cc/2025/Conference — NeurIPS 2025 spotlight_

### Official Review · Reviewer_9riX · 2025-06-14

**Clarity:** 2
**Significance:** 3
**Originality:** 3
**Rating:** 4
**Confidence:** 4

**Summary:**

The paper challenges the common assumption that longer input sequences always benefit time series forecasting. To this end, it introduces two techniques aimed at improving both accuracy and efficiency by filtering out irrelevant information: Adaptive Masking Loss (AML), which suppresses temporal prefixes that contribute little to predictions, and Embedding Similarity Penalty (ESP), which promotes semantic alignment between input embeddings and outputs to enhance generalization. The proposed methods are tested across several datasets (ETTh1, ETTh2, Solar-Energy, Weather) and architectures, including Transformers and MLP-based models. The authors provide ablation studies that highlight the contributions of AML and ESP individually and in combination. The results show consistent performance improvements and point to new directions for optimization in time series forecasting.

**Questions:**

Please address the weaknesses.

**Ethical Concerns:**

["NO or VERY MINOR ethics concerns only"]

**Final Justification:**

The authors have thoroughly addressed the concerns raised during the review and discussion process. In light of their clarifications and additional analyses, I have adjusted my score upward to reflect a more positive evaluation.

**Limitations:**

yes

**Paper Formatting Concerns:**

No major issues

**Quality:**

3

**Strengths And Weaknesses:**

Strengths
* Both AML and ESP are conceptually simple and easy to grasp.
* The methods consistently improve performance across various models and datasets, with solid experimental backing.

Weaknesses
* The core motivation questioning the value of longer input sequences may seem at odds with prior work, where longer sequences often help by providing additional useful context. While long inputs can certainly introduce noise, this is a well-studied challenge in time series forecasting, and recent studies (e.g., TimeMoE) have demonstrated clear performance gains with longer sequences. The authors could clarify their framing to better align with or differentiate from these established findings.

* The connection to existing work on noise robustness and time series representation learning feels underdeveloped. The paper focuses mainly on applying the proposed methods to baseline models, but it would benefit from a clearer discussion of how it builds on or differs from related approaches.

* The claim that AML’s stochastic sampling reduces computational cost isn’t substantiated with quantitative evidence. There’s no analysis of how varying the sample size affects training time, resource usage, or accuracy.

* While the ablation results show that ESP complements AML, the paper doesn’t provide insight into why ESP is particularly helpful for certain datasets or tasks.

* There’s little analysis of the kinds of patterns AML tends to mask, whether it targets noise, outliers, or outdated information is left unclear.

* The argument that optimal mask positions mostly fall in prefix regions isn’t supported by quantitative or visual analyses (e.g., distributions of k, per dataset and model).

* Some recent baseline models and datasets are missing or feel outdated (e.g., Traffic, ExchangeRate, Illness), limiting the scope of comparison.

---

> ### Author Rebuttal · Authors · 2025-07-31
>
> Dear Reviewer:
>
> Thank you for your careful and in-depth review. From the perspectives of theoretical rigor, completeness of experimental design, and the connection between our methodological innovations and existing research, your suggestions are highly constructive and crucial to improving this paper. The core tensions you identified have all clarified the key directions for revision. We will address these issues one by one below.
>
> **A1：** We are deeply grateful for your insightful comments. Existing research—including strong work such as **TimeMoE**—indeed shows that longer sequences can improve model performance by providing richer contextual information. It may be that we did not convey this clearly; we fully agree with this view and have built upon it with further exploration.
>
> Our study focuses on the **selective utilization** of information within long sequences. While longer sequences do increase the total amount of information, our experiments indicate that some portions may be unhelpful for the prediction target or even introduce interference (which aligns with classic findings on noise in time-series forecasting). To address this, our masking mechanism aims to more effectively filter and leverage key subsequences, dynamically adjusting via the parameter $\beta$ (defined as $\beta = \max\left(0, \frac{\ell - \ell_s^{*}}{\ell}\right)$). This allows us to increase the information density of the learned representation without degrading predictive performance.
>
> Following the evaluation protocol in Section 2, we applied an optimal masking analysis to a trained TimeMoE model (official GitHub code and hyperparameters) to diagnose redundancy. Due to time constraints, we evaluated the single-expert configuration on ETTh1 with input length L=1024 and forecast horizons of 96 and 192. As shown in Table following, while longer inputs provide richer context, they also introduce substantial redundant features（ratio denotes the proportion of samples in the dataset whose prediction MSE decreases after applying the  optimal mask）.
>
> | Prediction length | Dataset | Original loss | Masked loss |   Ratio  |
> |:-----------------:|:-------:|--------------:|------------:|---------:|
> |        96         | train   |         0.106 |       0.096 |  99.13%  |
> |        96         | test    |         0.522 |       0.463 |  98.24%  |
> |       192         | train   |         0.157 |       0.149 |  98.67%  |
> |       192         | test    |         0.722 |       0.674 | 100.00%  |
>
> **A2：** Thank you very much for the suggestion. Prior to our work, on **noise robustness**, Informer [1] observed that as input sequences grow longer, the computational complexity of traditional models increases, which can degrade forecasting performance; it addresses this via sparse attention for information distillation. For **representation learning**, TS2Vec [2] adopts contrastive learning to regularize temporal representations. However,most long-sequence forecasting studies emphasize architecture design, whereas our paper approaches the problem from the perspective of objective optimization.
>
> The distinction between AML and prior noise-robustness research is that AML uses information-theoretic guidance to eliminate redundant information. The contribution of ESPto representation learning is to constrain input–output consistency. Due to space limitations in the main text, we will add these comparisons to the Related Work section of the camera-ready supplementary material.
>
>
> 【1】Zhou H, Zhang S, Peng J, et al. Informer: Beyond efficient transformer for long sequence time-series forecasting[C]//Proceedings of the AAAI conference on artificial intelligence. 2021, 35(12): 11106-11115.
>
> 【2】Yue Z, Wang Y, Duan J, et al. Ts2vec: Towards universal representation of time series[C]//Proceedings of the AAAI conference on artificial intelligence. 2022, 36(8): 8980-8987.
>
> **A3：** Thank you for pointing this out; we did not state it clearly in the original manuscript. The AML method uses a random-sampling strategy that effectively reduces computational cost, striking a balance between efficiency and performance. Compared with the high complexity of exhaustively evaluating all possible masks (time complexity $O(L,n)$), fixing the sample to $m$ masks reduces the complexity to $O(m,n)$. We analyzed how the sampling size affects training time, resource usage, and accuracy, and found that although the loss decreases as $m$ increases, the rate of decrease diminishes. We therefore set $m=12$ as a trade-off between computational cost and efficiency.
>
>  Here, $m$ is the number of masks and $L$ is the input length; “MSE (mean)” and “MAE (mean)” are averaged over different output lengths.
>
>
> | Model        |  m/l | MSE (mean) | MAE (mean) |
> |:-------------|:----:|-----------:|-----------:|
> | SOFTS        |  1/8 |      0.407 |      0.416 |
> | SOFTS        |  1/6 |      0.407 |      0.409 |
> | SOFTS        |  1/4 |      0.389 |      0.393 |
> | SOFTS        |  1/3 |      0.384 |      0.388 |
> | SOFTS        |  1/2 |      0.381 |      0.385 |
> | SOFTS        |  3/4 |      0.380 |      0.384 |
> | iTransformer |  1/8 |      0.412 |      0.412 |
> | iTransformer |  1/6 |      0.411 |      0.417 |
> | iTransformer |  1/4 |      0.402 |      0.399 |
> | iTransformer |  1/3 |      0.398 |      0.395 |
> | iTransformer |  1/2 |      0.395 |      0.392 |
> | iTransformer |  3/4 |      0.394 |      0.390 |
>
> **A4：** Thank you for the insightful comments. We observe heterogeneous performance of ESP across datasets of different dimensionalities: ESP performs better on ETTh1, ETTh2, ETTm1, and ETTm2, while the gains are smaller on higher-dimensional datasets such as Weather (21 channels) and Solar-Energy (137 channels). The main reason is that ESP enhances robustness by explicitly aligning the geometric structure of the input embedding space with that of the output space, directly constraining pairwise relative distances between samples; as the number of channels (feature dimensionality) increases, the optimization directions grow exponentially with dimensionality, introducing greater uncertainty during training and potentially yielding smaller improvements than on lower-dimensional data. However, high-dimensional datasets typically contain more redundant information, on which AML performs well; combining ESP with AML therefore yields consistent and effective improvements across diverse datasets.
>
>
> **A5:** Thank you very much for the valuable comments. The issue you raised regarding the types of mask patterns in AML is indeed important and helps improve the interpretability of our method. AML dynamically adjusts the optimization strength via an adaptive weight $\beta$, so mask variants that yield larger reductions in loss exert a stronger influence on the training objective, guiding the model to learn features with less redundancy. We fully understand the reviewer’s interest in clearly distinguishing the types of feature patterns being masked. Although the current masking strategy improves representation efficiency, it does have limitations in interpretability: while the adaptive-weight mechanism is effective, its decision process remains something of a black box, making it difficult to clearly identify the specific types of masked feature patterns (e.g., noise, outliers, or stale information). We take this concern seriously and consider it an important direction for future work.
>
> **A6:** Thank you for the valuable comments. We acknowledge that searching for the optimal mask over the interval $[K, K+n]$ may outperform restricting the mask position to the prefix region. However, under our current search procedure, retrieval over $[K, K+n]$ is prohibitively expensive. Our focus on the prefix region is motivated by the Markov chain assumption: many real-world time series (e.g., traffic flow, energy consumption) exhibit Markov properties, where future states depend primarily on recent history and the contribution of distant history decays with temporal distance. Accordingly, we posit that later segments already contain the information present in the prefix, making redundancy more prevalent in the prefix. Once we have improved the retrieval method, we plan to expand the search range to $[K, K+n]$.
>
> **A7：** We sincerely appreciate the reviewer’s valuable suggestions. The benchmark datasets you mentioned—Traffic, ExchangeRate, and Illness—are indeed important standards in time-series forecasting. In designing our experiments, we primarily considered two factors: (1) selecting public datasets consistent with those used in the original baseline papers to ensure comparability; and (2) adopting the officially released hyperparameter settings to ensure fairness in reproduction (especially given the sensitivity of time-series models to hyperparameters). Accordingly, we used, wherever possible, the intersection of datasets for which the baseline models provide official hyperparameters.
>
> Regarding the benchmarking issue you raised, we encountered several practical difficulties. For the Traffic dataset, the large data scale and the already high computational complexity of the baseline models made testing challenging under limited compute, especially since our method introduces additional spatiotemporal complexity. In addition, for the ExchangeRate and Illness datasets, some baselines do not publicly release hyperparameter settings, which complicates fair comparison. If you deem it necessary, we are willing to conduct supplemental experiments on these datasets; however, we note that, due to hyperparameter uncertainty, the results may differ from the baselines’ performance reported on other datasets.
>
> Meanwhile, we have been actively tracking the latest developments in the field; for example, SOFTS is a NeurIPS 2024 paper. If there are recent open-source models of interest, we would be happy to include them in our comparisons. Your professional advice will greatly help us improve this work.

---

> ### Comment · Reviewer_9riX · 2025-08-05
>
> Thank you for the detailed response. While some of my concerns have been addressed, the response is not entirely satisfying.
>
> Both Informer and TS2Vec are relatively older works. Although they made notable contributions at the time, there have been significant advances in the field since their publication. Moreover, these models were not primarily designed for noise robustness, which makes them less ideal for evaluating that aspect.
>
> The notion of redundancy in the paper also feels quite abstract. Without clearer specification, it may come across as overly interpreted and could weaken the methodological justification. In time-series tasks where performance improvements are often marginal, interpretability and a well-grounded rationale are especially important to demonstrate practical advantages.
>
> I understand that benchmarking across multiple models and domains involves substantial overhead. I do not expect the proposed method to outperform all baselines in every domain, as that would be unrealistic. However, comparative results, even on a limited scale, help readers understand how the method relates to existing work. They also provide confidence that the authors are not selectively reporting only favorable outcomes. Even if some prior models do not disclose full hyperparameter details, reasonable tuning, particularly on smaller datasets like ExchangeRate or Illness, should still be feasible. The response on this point felt somewhat unconvincing.
>
> I hope the authors consider improving these aspects.

---

> > ### Author Response · Authors · 2025-08-06
> >
> > **A1:** Thank you for your correction. We fully understand the reviewer’s reasonable concern regarding the use of earlier models (e.g., Informer and TS2Vec) in the related work section. Indeed, in recent years, there have been many new research advances in noise robustness and time series representation learning, including directions highly aligned with our research objectives. For example, Zhang et al. proposed the TS-CoT framework (CIKM 2023) [1], which employs a dual-encoder architecture to construct complementary views, achieving global semantic consistency learning through a cross-view prototype alignment mechanism. By incorporating a sliding average prototype update strategy, TS-CoT enhances robust representation learning under unsupervised/semi-supervised conditions and effectively mitigates noise interference. Similarly, Zhou et al. proposed the DECL method (IJCAI 2024) [2], which constructs positive samples from denoised data and negative samples by adding noise, guiding contrastive learning to acquire denoising capabilities. Furthermore, DECL adopts a reconstruction-error-driven adaptive denoiser selection strategy, achieving end-to-end noise-robust representation learning.
> >
> > Compared with the above approaches, our proposed AML does not require modifications to the encoder architecture or the construction of contrastive samples. Instead, from an information-theoretic perspective, AML adaptively discovers the most predictive masked representations directly from the sequence data itself and uses them as optimization targets to guide the model in suppressing redundant feature learning. This enables AML to constrain redundant information internally within the model in a simpler manner, thereby improving representation compactness and noise robustness without increasing the structural complexity of the model.
> >
> > [1] Zhang W, Zhang J, Li J, et al. A co-training approach for noisy time series learning [C] // Proceedings of the 32nd ACM International Conference on Information and Knowledge Management. 2023: 3308–3318.
> > [2] Zhou S, Zha D, Shen X, et al. Denoising-aware contrastive learning for noisy time series. Proceedings of the Thirty-Third International Joint Conference on Artificial Intelligence (IJCAI-24), 2024.
> >
> > It is worth noting that TS-CoT (CIKM 2023) does not explicitly define “noise”; DECL (IJCAI 2024) also does not provide a clearer definition, merely describing it as “many time series (e.g., bio-signals collected from sensors) naturally suffer from noises that can severely change the data characteristics and impair representation learned by SSL algorithms.” In these papers, “noise” is described as task-irrelevant or harmful information, which is conceptually similar to what we refer to as “redundant information” in our work.
> >
> > Moreover, these methods do not theoretically explain how such redundant (or noisy) information affects model representations, nor do they offer interpretable mechanisms for balancing “information retention” and “task relevance” in representation learning. The interpretability of denoising remains an important open challenge in the time series domain. We also recognize the importance of interpretability in time series analysis; although it is not the primary focus of our paper, we still provide a detailed description of some noise-induced anomalies in Section 2. Further exploration and breakthroughs in this area are left for future work.
> >
> > ---
> >
> > **A2:** Thank you very much for your suggestion. However, regarding your concern about “selective reporting of results,” we can state very seriously that this was not the case. The reason we did not initially include experiments on the ExchangeRate and Illness datasets, as mentioned in our first reply, was due to considerations of fairness. Nevertheless, since you are highly interested in the results on these two datasets, we are more than happy to supplement this part of the experiments. Given current time constraints, we are conducting related tests on computationally efficient models (e.g., SOFTS) to obtain preliminary results as quickly as possible. We will do our best to supplement these results on SOFTS before the end of the discussion phase to further enhance the completeness and persuasiveness of the work.

---

> ### Author Response · Authors · 2025-08-08
>
> Thank you for your valuable suggestion. We have conducted comprehensive experiments with SOFTS on the illness and ExchangeRate datasets, using 10 different random seeds, followed by significance testing. The results are summarized as follows:
>
> For the illness dataset, due to its extremely limited number of data points, our original setting (input sequence length L = 48, output sequence length $ H \in \{48, 72, 96, 120, 144, 168, 192\} $ ) was not feasible. Following the experimental setup of PatchTST, we adjusted the input length $ L $  to 48 and set the output lengths to $ H \in { 24, 36, 48, 60 } $ .  Conf MSE/Conf MAE indicates the confidence level for significance testing.
>
> **illness Dataset**
>
> |   H    | **SOFTS+AMRC MSE** | **SOFTS+AMRC MAE** | **SOFTS MSE** | **SOFTS MAE** | Conf-MSE | Conf-MAE |
> | :----: | :----------------: | :----------------: | :-----------: | :-----------: | :------: | :------: |
> | **24** |    1.633 ± 0.07    |    0.789 ± 0.06    | 1.776 ± 0.14  | 0.852 ± 0.03  |    95    |    95    |
> | **36** |    1.858 ± 0.07    |    0.854 ± 0.06    | 1.942 ± 0.12  | 0.904 ± 0.04  |    90    |    95    |
> | **48** |    2.035 ± 0.07    |    0.916 ± 0.05    | 2.153 ± 0.12  | 0.954 ± 0.03  |    95    |    95    |
> | **60** |    2.054 ± 0.07    |    0.935 ± 0.03    | 2.113 ± 0.10  | 0.958 ± 0.03  |    90    |    90    |
>
> **ExchangeRate Dataset**
>
> |    H    | **SOFTS+AMRC MSE** | **SOFTS+AMRC MAE** |  **SOFTS MSE**  |  **SOFTS MAE**   | Conf MSE | Conf MAE |
> | :-----: | :----------------: | :----------------: | :-------------: | :--------------: | :------: | :------: |
> | **48**  |  0.03913 ± 0.001   |  0.13016 ± 0.007   | 0.04208 ± 0.001 | 0.13728 ± 0.008  |    99    |    95    |
> | **72**  |  0.05788 ± 0.002   |  0.16432 ± 0.008   | 0.06093 ± 0.003 | 0.17116 ± 0.009  |    95    |    90    |
> | **96**  |  0.07927 ± 0.002   |  0.18782 ± 0.005   | 0.08329 ± 0.004 | 0.20196 ± 0.001 |    95    |    99    |
> | **120** |  0.10053 ± 0.001   |  0.21698 ± 0.002   | 0.10695 ± 0.001 | 0.22840 ± 0.001  |    99    |    99    |
> | **144** |  0.12417 ± 0.002   |  0.24484 ± 0.002   | 0.12935 ± 0.002 | 0.25241 ± 0.001  |    99    |    99    |
> | **168** |  0.14602 ± 0.002   |  0.25976 ± 0.001   | 0.16047 ± 0.003 | 0.28234 ± 0.003  |    99    |    99    |
> | **192** |  0.17457 ± 0.007   |  0.29181 ± 0.006   | 0.18376 ± 0.001 | 0.30397 ± 0.007  |    95    |    99    |
>
> In this case, the improvement margin and statistical significance on the illness dataset were less pronounced compared to other datasets. Upon closer examination, we found that illness contains only 967 data points in total.
>
> In contrast, the ExchangeRate dataset showed significance levels and improvement magnitudes consistent with those observed in other datasets reported in our paper. If you would like, we can provide the detailed results from all 10 runs along with the corresponding p-values from the significance tests, and include them. directly in the Discussion section.

---

> > ### Comment · Reviewer_9riX · 2025-08-08
> >
> > Thank the authors for their thoughtful response. While the degree of statistical significance differs, it is interesting to observe consistent improvements across both datasets. Given that these datasets are known to be challenging even for existing time-series forecasting methods, I believe these results are valuable and warrant further exploration.
> >
> > As mentioned earlier, I place great importance on establishing connections with prior work, and I am glad to see that the authors have taken this aspect into account.
> >
> > My main concerns have been resolved.

---

### Official Review · Reviewer_mCJD · 2025-07-02

**Clarity:** 4
**Significance:** 3
**Originality:** 3
**Rating:** 5
**Confidence:** 4

**Summary:**

The authors propose Adaptive Masking Loss with Representation Consistency (AMRC), a training scheme for time-series models that draws on information-bottleneck theory. It introduces a dynamic masking loss that automatically highlights the most informative temporal segments during training, steering gradient updates away from redundant patterns, and it adds a representation-consistency constraint to keep the relationship among inputs, labels, and predictions stable. By jointly reducing noise and preserving meaningful structure in the learned features, AMRC improves forecast accuracy and demonstrates a more efficient, principled way to build robust temporal models.

**Questions:**

Q1. It seems like once masks are learned for a given task there could be some additional optimization to increase inference efficiency. Even if there was no performance gain, it would be helpful for storage and memory reasons to reduce the length of the sequence of any given time series problem. Is there any work the authors have explored to analyze what these efficiency wins may be under different task settings?

Q2. In some previous works related to time series interpretability there has been analysis of applying a dynamic mask [https://arxiv.org/pdf/2106.05303] or sliding window [https://arxiv.org/pdf/2107.14317] to analyze how predictions change. Would further gains be possible if the masking operator was defined for some sequence from k to k+n rather than just from time 0:k.

**Ethical Concerns:**

["NO or VERY MINOR ethics concerns only"]

**Final Justification:**

The authors have made a great effort to further examine the reviews of all reviewers and provide additional experimentation where helpful to validate their results. I am convinced by the sensitivity analysis and comparisons with alternative methods that were further added to the current results and will keep my recommendation as is.

**Limitations:**

Yes

**Paper Formatting Concerns:**

Typo: caption for table 2 says AMRC model results are highlighted in bold where the model improved by more than ‘0.05’ but it seems it should be ‘0.005’

**Quality:**

4

**Strengths And Weaknesses:**

S1. The core idea of the paper is interesting and novel. While there has been work on improving interpretability of time series predictions through the application of IB-inspired masking this hasn’t been applied directly to improving the underlying learned representation to the best of my knowledge. Other methods such as TimeSieve [https://arxiv.org/pdf/2406.05036] have only recently started exploring how to improve the underlying representations in an efficient way.

S2. Framework flexibility: the authors do show some performance gains on a range of model types and tasks. The design of the framework is such that it should be easy to extend as needed in many places.

S3. Ablation studies were useful for understanding the impact of the ESP step combined with AML as well as how the method scales across tasks and model types when considering what % of training samples show improved performance after masking. These studies suggest the method is effective in its aims of reducing redundant feature learning.

W1. It is clear that certain tasks and data types will benefit more from this method than others. It would be helpful to provide guidance on how to identify these cases where the potential performance gain may outway the efficiency costs of introducing a more heavyweight method like this one.

---

> ### Author Rebuttal · Authors · 2025-07-31
>
> Dear Reviewer:
>
> We sincerely appreciate your insightful and constructive comments on our manuscript, and we are especially grateful for your careful identification of typographical errors, which we have corrected in the revised manuscript. Your professional review has greatly helped us improve the quality of the paper. In response to your questions and suggestions, we have made thorough revisions and added clarifications; our detailed replies are as follows:
>
> **Q1:**  It seems like once masks are learned for a given task there could be some additional optimization to increase inference efficiency. Even if there was no performance gain, it would be helpful for storage and memory reasons to reduce the length of the sequence of any given time series problem. Is there any work the authors have explored to analyze what these efficiency wins may be under different task settings?
>
> **A1:** Thank you for raising this important question. We have indeed thought carefully about efficiency optimizations at inference time. Building on AML’s core idea—improving representation quality by masking out redundant information in the input sequence—we find that while the current random-masking strategy improves upon exhaustive search (with complexity $O(L)$), there is still room for further optimization. Given that redundancy in real-world scenarios often exhibits continuity (e.g., prolonged repeated patterns or stable fluctuations), the following strategies may help improve efficiency:
>
> 1\) Dynamic binary search strategy: This method assumes redundancy is contiguous and models the mask-length space $[0, L]$ as an ordered search space. Through iterative binary search, it can quickly locate the contiguous interval containing redundant information: in each round, evaluating the performance of masks at two midpoints halves the search space. This method applies only when redundant features are contiguous.
>
> 2\) Gradient-based adaptive sampling: This method leverages AML’s loss-feedback mechanism. By analyzing the trend of the loss across neighboring mask lengths, it dynamically adjusts the sampling direction. When a significant drop in loss is detected, it indicates that the current mask length is closer to the optimal de-redundancy region; the procedure automatically shifts in that direction, achieving a hill-climbing search. This method adapts to different data distributions and is also effective for non-contiguous redundancy.
>
> Future work can further explore combining masking with retrieval to achieve more precise redundancy removal and more efficient representation learning. Regarding efficiency gains across different scenarios, our method performs well in most settings. However, when predictive signals are concentrated very early in the timeline, AML’s prefix-masking policy (starting at $0$) may, after multiple masking evaluations, yield a penalty coefficient $\beta = \max\left(0, \frac{\ell - \ell_s^{*}}{\ell}\right)$ that remains identically zero, adding extra spatiotemporal cost without improving performance. In addition, ESP shows relatively limited gains on high-dimensional datasets (e.g., Weather and Solar-Energy) because ESP is based on relative distances and explicitly aligns the geometry of the input embedding space with that of the output space. For datasets with many channels, a fixed set of relative distances leaves more degrees of freedom, making distance-metric optimization more challenging. For example, in 2D, knowing a point’s relative distances to two other points can determine its position, whereas in 3D one needs three points.
>
> **Q2：** In some previous works related to time series interpretability there has been analysis of applying a dynamic mask [https://arxiv.org/pdf/2106.05303] or sliding window [https://arxiv.org/pdf/2107.14317] to analyze how predictions change. Would further gains be possible if the masking operator was defined for some sequence from k to k+n rather than just from time 0:k.
>
> **A2:** Thank you very much for the suggestion. Your proposal to mask the interval $[k,k+n]$ is indeed more advantageous, and we agree it is preferable. However, due to computational constraints, our current choice of a prefix mask $[0,k]$ (i.e., $0{:}k$) is based on two considerations: (1) exhaustive search over $[k,k+n]$ leads to exponentially growing computational complexity, which is prohibitive with our current unoptimized retrieval procedure; (2) many time-series settings adopt a Markov assumption, under which later segments implicitly encode earlier information, so we temporarily mask the earlier segment. Exploring a joint masking–retrieval approach is a promising direction for future work—thank you for highlighting this important direction.

---

> > ### Comment · Reviewer_mCJD · 2025-08-06
> >
> > Thank you for preparing the responses.
> >
> > It seems that some combination of adaptive sampling and perhaps even additional sampling over a range of mask choices depending on the style of the task could lead to further performance and optimization of the method. I understand that there is a significant increase in the optimization space by moving to a sliding window approach, I wonder if it would be possible to see the ablation studies shared in the response to reviewer nsY4 but instead of masking up to m/l = 3/4 from [0:k] to also evaluate from [n-k:n]?
> >
> > I appreciate the suggestions of both dynamic binary search and gradient-based adaptive sampling. It seems this could be direction for further research and is not necessary to specifically improve the current paper as is.
> >
> > All that said, this does make me question whether there might be some naive heuristic of masking that would show relative gains for less computational cost. Have the authors tried just masking the first half of all sequences as another ablation to compare against the individual AML module?
> >
> > Finally in comparing these methods, it is unclear how statistically significant the difference in performance is between the AMRC and original metrics. Have the authors conducted K-fold cross-validation or tests for statistical significance for AMRC? I believe previous works like TimeMixer++ showed this kind of rigour.

---

> > > ### Author Response · Authors · 2025-08-06
> > >
> > > We sincerely appreciate your recognition of our work and your constructive feedback. Your suggestions are of great importance to improving this study.
> > >
> > > **A1:** Regarding your suggestion to further evaluate the masking strategy applied to the [n−k:n] interval, since you expressed interest in this, we are more than happy to include this verification. Due to time constraints, we are currently prioritizing this experiment on the most computationally efficient model (SOFTS) so that we can obtain preliminary results as quickly as possible. We will make every effort to update and supplement the corresponding evaluation results before the end of the discussion phase.
> > >
> > > ---
> > >
> > > **A2:** On your question of whether there could be a simpler heuristic masking method, we did in fact conduct similar attempts in our early experiments. The initial inspiration for our method came from a test on the SOFTS model: during inference, we fixed the mask on the first 6 time steps and observed an average MSE reduction of about 0.001, which was a pleasant surprise. However, further experiments showed that this strategy was unstable. For example, applying the same operation (masking the first 6 time steps) on PatchTST led to an average MSE increase of about 0.005. Interestingly, masking the first 12 time steps in PatchTST reduced the average MSE by about 0.003.
> > >
> > > For this reason, we did not include the heuristic approach of “uniformly masking the first half of the sequence” as a baseline in the paper—it is overly simplistic and its performance is highly inconsistent. We were unable to identify a reliable pattern, such as a fixed starting point or masking length, that consistently yielded good results. That said, if you believe such a baseline would help illustrate our work, we would be glad to include it in the main text or appendix as supplementary material. Please feel free to let us know, and we can present these results directly in the discussion section.

---

> > > > ### Author Response · Authors · 2025-08-06
> > > >
> > > > **A3:** Regarding your final point, in Appendix E (Table 5), we originally presented the mean results from ten runs under different input lengths, each run using a different random seed. We initially believed this presentation sufficiently conveyed the significance of performance differences. However, based on your suggestion, we realized that the current format could be clearer. Following your advice, we have reformatted the results as mean ± standard deviation and conducted significance tests. The updated table is as follows:
> > > >
> > > > Both Conf MSE and Conf MAE denote confidence levels
> > > >
> > > > | Dataset      | AMRC + TSMixer&nbsp;MSE | AMRC + TSMixer&nbsp;MAE | TSMixer&nbsp;MSE | TSMixer&nbsp;MAE | Conf&nbsp;MSE（%） | Conf&nbsp;MAE（%） |
> > > > | :----------- | :---------------------- | :---------------------- | :--------------- | :--------------- | :----------------: | :----------------: |
> > > > | ETTh1        | 0.386 ± 0.010           | 0.397 ± 0.008           | 0.402 ± 0.003    | 0.412 ± 0.005    |         99         |         99         |
> > > > | ETTh2        | 0.319 ± 0.007           | 0.340 ± 0.011           | 0.324 ± 0.004    | 0.357 ± 0.004    |         99         |         99         |
> > > > | ETTm1        | 0.432 ± 0.010           | 0.412 ± 0.006           | 0.440 ± 0.003    | 0.413 ± 0.006    |         99         |         95         |
> > > > | ETTm2        | 0.196 ± 0.007           | 0.257 ± 0.013           | 0.201 ± 0.005    | 0.279 ± 0.003    |         95         |         99         |
> > > > | Solar-Energy | 0.280 ± 0.011           | 0.313 ± 0.005           | 0.288 ± 0.006    | 0.314 ± 0.004    |         95         |         99         |
> > > > | Weather      | 0.212 ± 0.010           | 0.281 ± 0.009           | 0.222 ± 0.002    | 0.288 ± 0.007    |         99         |         99         |
> > > > | Electricity  | 0.169 ± 0.009           | 0.247 ± 0.006           | 0.173 ± 0.005    | 0.258 ± 0.006    |         99         |         95         |
> > > >
> > > > | Dataset      | AMRC + TimeMixer&nbsp;MSE | AMRC + TimeMixer&nbsp;MAE | TimeMixer&nbsp;MSE | TimeMixer&nbsp;MAE | Conf&nbsp;MSE（%） | Conf&nbsp;MAE（%） |
> > > > | :----------- | :------------------------ | :------------------------ | :----------------- | :----------------- | :----------------: | :----------------: |
> > > > | ETTh1        | 0.388 ± 0.006             | 0.401 ± 0.007             | 0.393 ± 0.003      | 0.408 ± 0.005      |         99         |         99         |
> > > > | ETTh2        | 0.316 ± 0.008             | 0.339 ± 0.010             | 0.318 ± 0.006      | 0.355 ± 0.008      |         95         |         99         |
> > > > | ETTm1        | 0.447 ± 0.008             | 0.405 ± 0.009             | 0.466 ± 0.004      | 0.429 ± 0.006      |         99         |         99         |
> > > > | ETTm2        | 0.204 ± 0.007             | 0.269 ± 0.006             | 0.209 ± 0.002      | 0.285 ± 0.004      |         95         |         99         |
> > > > | Solar-Energy | 0.284 ± 0.008             | 0.317 ± 0.001             | 0.288 ± 0.003      | 0.317 ± 0.001      |         95         |         90         |
> > > > | Weather      | 0.186 ± 0.014             | 0.228 ± 0.011             | 0.197 ± 0.010      | 0.237 ± 0.009      |         99         |         99         |
> > > > | ECL          | 0.188 ± 0.012             | 0.277 ± 0.008             | 0.194 ± 0.010      | 0.279 ± 0.006      |         99         |         95         |
> > > >
> > > > | Dataset      | AMRC + iTransformer&nbsp;MSE | AMRC + iTransformer&nbsp;MAE | iTransformer&nbsp;MSE | iTransformer&nbsp;MAE | Conf&nbsp;MSE（%） | Conf&nbsp;MAE（%） |
> > > > | :----------- | :--------------------------- | :--------------------------- | :-------------------- | :-------------------- | :----------------: | :----------------: |
> > > > | ETTh1        | 0.402 ± 0.004                | 0.399 ± 0.005                | 0.413 ± 0.001         | 0.415 ± 0.002         |         99         |         99         |
> > > > | ETTh2        | 0.324 ± 0.005                | 0.356 ± 0.004                | 0.329 ± 0.002         | 0.362 ± 0.002         |         95         |         99         |
> > > > | ETTm1        | 0.502 ± 0.004                | 0.447 ± 0.002                | 0.517 ± 0.003         | 0.448 ± 0.001         |         99         |         99         |
> > > > | ETTm2        | 0.211 ± 0.003                | 0.280 ± 0.003                | 0.213 ± 0.001         | 0.290 ± 0.002         |         95         |         99         |
> > > > | Solar-Energy | 0.392 ± 0.006                | 0.342 ± 0.005                | 0.395 ± 0.002         | 0.352 ± 0.002         |         95         |         99         |
> > > > | Weather      | 0.201 ± 0.005                | 0.221 ± 0.008                | 0.209 ± 0.003         | 0.237 ± 0.002         |         99         |         99         |
> > > > | Electricity  | 0.163 ± 0.004                | 0.239 ± 0.007                | 0.176 ± 0.002         | 0.260 ± 0.003         |         99         |         99         |

---

> > > > > ### Author Response · Authors · 2025-08-06
> > > > >
> > > > > | Dataset      | PatchTST + AMRC&nbsp;MSE | PatchTST + AMRC&nbsp;MAE | PatchTST&nbsp;MSE | PatchTST&nbsp;MAE | Conf&nbsp;MSE（%） | Conf&nbsp;MAE（%） |
> > > > > | :----------- | :----------------------- | :----------------------- | :---------------- | :---------------- | :----------------: | :----------------: |
> > > > > | ETTh1        | 0.411 ± 0.005            | 0.415 ± 0.003            | 0.424 ± 0.003     | 0.424 ± 0.002     |         99         |         99         |
> > > > > | ETTh2        | 0.319 ± 0.004            | 0.356 ± 0.004            | 0.327 ± 0.001     | 0.358 ± 0.003     |         99         |         95         |
> > > > > | ETTm1        | 0.456 ± 0.004            | 0.413 ± 0.003            | 0.461 ± 0.003     | 0.422 ± 0.002     |         99         |         99         |
> > > > > | ETTm2        | 0.196 ± 0.005            | 0.271 ± 0.004            | 0.211 ± 0.002     | 0.287 ± 0.003     |         99         |         99         |
> > > > > | Solar-Energy | 0.361 ± 0.006            | 0.376 ± 0.007            | 0.374 ± 0.003     | 0.383 ± 0.004     |         95         |         99         |
> > > > > | Weather      | 0.210 ± 0.003            | 0.264 ± 0.003            | 0.215 ± 0.002     | 0.280 ± 0.003     |         99         |         99         |
> > > > > | Electricity  | 0.207 ± 0.004            | 0.285 ± 0.002            | 0.211 ± 0.002     | 0.283 ± 0.002     |         99         |         95         |
> > > > >
> > > > > | Dataset      | PatchTST + AMRC&nbsp;MSE | PatchTST + AMRC&nbsp;MAE | PatchTST&nbsp;MSE | PatchTST&nbsp;MAE | Conf&nbsp;MSE（%） | Conf&nbsp;MAE（%） |
> > > > > | :----------- | :----------------------- | :----------------------- | :---------------- | :---------------- | :----------------: | :----------------: |
> > > > > | ETTh1        | 0.389 ± 0.011            | 0.393 ± 0.009            | 0.408 ± 0.004     | 0.414 ± 0.003     |         99         |         99         |
> > > > > | ETTh2        | 0.311 ± 0.008            | 0.362 ± 0.004            | 0.326 ± 0.003     | 0.359 ± 0.004     |         99         |         95         |
> > > > > | ETTm1        | 0.475 ± 0.006            | 0.423 ± 0.005            | 0.484 ± 0.004     | 0.434 ± 0.003     |         99         |         99         |
> > > > > | ETTm2        | 0.198 ± 0.007            | 0.265 ± 0.006            | 0.210 ± 0.002     | 0.285 ± 0.004     |         99         |         99         |
> > > > > | Solar-Energy | 0.290 ± 0.007            | 0.309 ± 0.007            | 0.293 ± 0.003     | 0.314 ± 0.004     |         95         |         95         |
> > > > > | Weather      | 0.196 ± 0.005            | 0.186 ± 0.004            | 0.205 ± 0.002     | 0.234 ± 0.003     |         99         |         99         |
> > > > > | Electricity  | 0.162 ± 0.006            | 0.244 ± 0.007            | 0.169 ± 0.003     | 0.255 ± 0.004     |         99         |         99         |
> > > > >
> > > > > If you would like more detailed information, please do not hesitate to tell us. We would be happy to provide the complete data from each run, including MAE, MSE, and the corresponding p-values from the significance tests.
> > > > >
> > > > > ---
> > > > >
> > > > > Once again, thank you for your careful review and valuable suggestions—they have been tremendously helpful in refining our work.

---

> > > > > > ### Comment · Reviewer_mCJD · 2025-08-07
> > > > > >
> > > > > > Thanks for the additional effort here.
> > > > > >
> > > > > > for A3, I think this makes sense and I appreciate the inclusion of the standard deviation and significance tests. I understand there are space constraints but including this in the appendix could be beneficial.
> > > > > >
> > > > > > With regards to A2 I find the negative result with regards to the stability of a simple heuristic approach quite compelling and how it led to your eventual proposal quite compelling. This seems worth highlighting in the main manuscript in someway (perhaps just a sentence with a reference to your results in the appendix as well. This seems helpful in further justifying the complexity of the approach.
> > > > > >
> > > > > > Finally for A1 I understand if it is not possible given my late reply, but I do think it could be a useful exploration for articulating the tradeoffs of the baseline choice to start with 0:k.
> > > > > >
> > > > > > For now I will keep my score as is and I appreciate the authors efforts to further improve the validity and robustness of their results.

---

> > > > > > > ### Author Response · Authors · 2025-08-07
> > > > > > > **Thank you for your suggestions**
> > > > > > >
> > > > > > > Thank you very much for your valuable suggestions. We will carefully incorporate them and revise the paper accordingly to further improve our work.

---

### Official Review · Reviewer_nsY4 · 2025-07-03

**Clarity:** 2
**Significance:** 3
**Originality:** 2
**Rating:** 5
**Confidence:** 3

**Summary:**

This paper investigates the phenomenon that longer historical data does not always lead to better accuracy in time series forecasting, primarily due to the model learning redundant or noisy features. To address this issue, the authors propose a simple, plug-and-play training framework called Adaptive Masking Loss with Representation Consistency (AMRC), which features two core components: 1) Dynamic masking loss and 2) Representation consistency constraint. AMRC dynamically masks potentially redundant temporal information during training, encouraging the model to focus on more informative segments and improving its generalization ability.

**Questions:**

Please check the weaknesses I mentioned earlier.

**Ethical Concerns:**

["NO or VERY MINOR ethics concerns only"]

**Final Justification:**

Upon revisiting the paper, other reviews and the authors’ responses, I acknowledge that while some concerns regarding robustness remain, in my opinion, the paper presents valuable technical contributions to the field and demonstrates strong potential. I appreciate the authors’ openness to further discussion and their commitment to improving the work. Overall, I now lean towards accepting the paper. I have updated the score from "Weak accept" to "Accept".

**Limitations:**

- The paper does not extensively discuss the robustness of the method to hyperparameter choices or provide guidelines for their selection.
- Limited discussion on potential scenarios where AMRC might be less effective or may introduce drawbacks, such as over-masking or hindering the learning of genuinely informative long-term dependencies.

**Paper Formatting Concerns:**

None.

**Quality:**

3

**Strengths And Weaknesses:**

Strengths:
- Apparently, this is an effective method to enhance time series forecasting by mitigating the negative effects of redundant feature learning.
- The authors evaluate AMRC across multiple datasets with varying characteristics and several baseline models. The consistent performance improvements validate the effectiveness and robustness of the approach.

Weaknesses:
- The method relies on masking strategies and hyperparameters (e.g., prefix length, mask sampling), but the paper lacks discussion on its robustness to these choices or guidance for tuning them.
- The writing quality needs improvement, and there are several inconsistencies throughout the manuscript, such as:
    * Inconsistent use of "Figure," "Fig.," and "Fig"
    * Lack of uniformity in font sizes across text elements within and between figures

---

> ### Author Rebuttal · Authors · 2025-07-31
>
> Dear Reviewer:
>
> We are sincerely grateful for your thorough review and valuable comments—especially your careful identification of writing-quality issues, which has been pivotal in improving the paper’s professionalism and readability. We have addressed the problems you noted in the revised manuscript. Your suggestions have not only strengthened the rigor of the paper but also clarified directions for methodological improvement. In accordance with your feedback, we have undertaken a comprehensive revision; our detailed point-by-point responses are as follows:
>
> **Q1:** The paper does not extensively discuss the robustness of the method to hyperparameter choices or provide guidelines for their selection.
>
> **A1:** You are correct that AMRC’s performance can be influenced by hyperparameter choices. We have added the following analysis:
>
> **Mask count m:** In principle, a larger m allows the model to search for the most informative segments over a longer history (expanding the window from K to K+n). However, under our current search procedure this enlarges the optimization space and increases computational cost. Our hyperparameter sensitivity study(The results are shown in the table below) shows that while the loss continues to decrease as \(m\) grows, the rate of decrease diminishes (i.e., diminishing returns). We therefore set \(m=12\) as a balance between computational cost and efficiency.
>
>  Here, *m* denotes the number of sampled masks and *L* is the input length; “MSE (mean)” and “MAE (mean)” are averaged across different output lengths.
>
> | Model        |  m/l | MSE (mean) | MAE (mean) |
> |:-------------|:----:|-----------:|-----------:|
> | SOFTS        |  1/8 |      0.407 |      0.416 |
> | SOFTS        |  1/6 |      0.407 |      0.409 |
> | SOFTS        |  1/4 |      0.389 |      0.393 |
> | SOFTS        |  1/3 |      0.384 |      0.388 |
> | SOFTS        |  1/2 |      0.381 |      0.385 |
> | SOFTS        |  3/4 |      0.380 |      0.384 |
> | iTransformer |  1/8 |      0.412 |      0.412 |
> | iTransformer |  1/6 |      0.411 |      0.417 |
> | iTransformer |  1/4 |      0.402 |      0.399 |
> | iTransformer |  1/3 |      0.398 |      0.395 |
> | iTransformer |  1/2 |      0.395 |      0.392 |
> | iTransformer |  3/4 |      0.394 |      0.390 |
> | TimeMixer    |  1/8 |      0.396 |      0.405 |
> | TimeMixer    |  1/6 |      0.391 |      0.409 |
> | TimeMixer    |  1/4 |      0.388 |      0.401 |
> | TimeMixer    |  1/3 |      0.385 |      0.398 |
> | TimeMixer    |  1/2 |      0.383 |      0.396 |
> | TimeMixer    |  3/4 |      0.382 |      0.395 |
> | PatchTST     |  1/8 |      0.422 |      0.420 |
> | PatchTST     |  1/6 |      0.419 |      0.421 |
> | PatchTST     |  1/4 |      0.411 |      0.415 |
> | PatchTST     |  1/3 |      0.406 |      0.411 |
> | PatchTST     |  1/2 |      0.403 |      0.409 |
> | PatchTST     |  3/4 |      0.401 |      0.408 |
> | TSMixer      |  1/8 |      0.401 |      0.411 |
> | TSMixer      |  1/6 |      0.395 |      0.408 |
> | TSMixer      |  1/4 |      0.386 |      0.397 |
> | TSMixer      |  1/3 |      0.381 |      0.392 |
> | TSMixer      |  1/2 |      0.378 |      0.389 |
> | TSMixer      |  3/4 |      0.376 |      0.387 |
>
> **Q2**:Limited discussion on potential scenarios where AMRC might be less effective or may introduce drawbacks, such as over-masking or hindering the learning of genuinely informative long-term dependencies.
>
> **A2**:Thank you for raising this key question. Our method does have limitations on some datasets. For AML, if the information most relevant to predicting $Y$ lies in the prefix time steps, AML’s prefix-masking policy (starting at 0) becomes suboptimal: multiple rounds of random masking increase computation, and the penalty coefficient $\beta = \max\left(0, \frac{\ell - \ell_s^{*}}{\ell}\right)$ remains identically zero, making performance gains unlikely. This highlights AML’s sensitivity to key information that does not lie in the prefix. For ESP, the improvements are relatively limited on high-dimensional datasets (e.g., Weather and Solar-Energy), because ESP enhances robustness by explicitly aligning the geometry of the input embedding space with that of the output space and directly constraining pairwise relative distances; as the number of channels (feature dimensionality) increases, the optimization directions grow exponentially with dimensionality, introducing greater uncertainty during optimization and potentially yielding smaller gains than on lower-dimensional datasets.

---

### Official Review · Reviewer_EQP6 · 2025-07-03

**Clarity:** 3
**Significance:** 3
**Originality:** 3
**Rating:** 5
**Confidence:** 3

**Summary:**

The authors make an interesting observation that recent NN architectures trained on standard benchmarks can improve their performance by masking the context windows, and that latent representation lack desirable properties. They leverage these empirical findings to construct representation penalizations to train time series models with better generalization. They demonstrate their approach in benchmark settings.

**Questions:**

1. Although the use of IB in representation learning may customary, it would be useful in section 2.3 to clarify the relationship between X, Y, Z and $\theta$
2. In 3.1 this can be said of any transformation applied to X that reduces entropy, to me the important part is the invariance or preservation of predictive power, or in other terms the mutual infomation between the label and the transformed input.
3. why does the first inequality in line 148 hold?
4. Is the ESP novel in the time-series literature - I didn't see any references on it.
5. The combination of AML and ESP is reminiscent of contrastive objectives in (self and semi-supervised) representation learning, have you explored connections?

**Ethical Concerns:**

["NO or VERY MINOR ethics concerns only"]

**Final Justification:**

I mantain my positive score, but increase my confidence in it thanks to the authors rebuttal clarifications.

**Limitations:**

Yes

**Quality:**

3

**Strengths And Weaknesses:**

Strengths:
* The empirical observation about performance of SOTA architectures with context masking is - to the best of my knowledge - novel, and of independent interest
* Augmenting time series reconstruction error objective with the proposed representation regularizations/losses  is also novel and well motivated
* Experimental results using standard benchmark settings and models demonstrate performance improvements
* Authors provide code

Weaknesses:
* Minor, some of the information theory connections/motivations/rationales are unclear to me in its current presentation.

---

> ### Author Rebuttal · Authors · 2025-07-31
>
> Thank you sincerely for your valuable comments. Your recognition of our empirical findings on context masking and so on is deeply encouraging. At the same time, the issues you identified—such as insufficient clarity in the theoretical exposition—provide important directions for further improving the work. We have carefully examined each of your points and made corresponding revisions (with changes in the manuscript). Below, we respond to your specific comments point by point, and we respectfully welcome any further guidance.
>
> **Q1**: Although the use of IB in representation learning may customary, it would be useful in section 2.3 to clarify the relationship between $X$, $Y$, $Z$ and $\theta$.
>
> **A1**: Thank you for your valuable suggestion! To more clearly clarify the relationships among $X$, $Y$, and $Z$, we will add the following clarification in Section 2.3: In time-series forecasting models, the input sequence $X$ is encoded into a latent representation $Z$, and the decoder then predicts the target $Y$ based on $Z$. The optimization objective is to learn an optimal representation $Z$ that maximally preserves the information relevant to $Y$. The informational relationships among $X$, $Y$, and $Z$ can be quantified using mutual information.
>
> **Q2**: In 3.1 this can be said of any transformation applied to X that reduces entropy, to me the important part is the invariance or preservation of predictive power, or in other terms the mutual infomation between the label and the transformed input.
>
> **A2**: You have pinpointed the core idea of our approach—eliminating redundant information without compromising predictive performance—which is precisely what motivates AML. Concretely, we introduce an adaptive coefficient $\beta = \max\left(0, \frac{\ell - \ell_s^{*}}{\ell}\right)$ and apply the information-bottleneck regularization only when the representation-learning loss $\ell_s$ is lower than the original prediction loss $\ell$ (i.e., $\ell_s < \ell$). This setup guarantees, by construction, that the transformed representation preserves the same predictive capability as the raw input. Your feedback helps us articulate the theoretical underpinnings more clearly. We will highlight this point more prominently in Section 3.1 of the revised manuscript.
>
> **Q3**: Why does the first inequality in line 148 hold?
>
> **A3**: What follows is a proof of this statement.
>
> **Assumptions.**
>
> - Original input: $X$
> - Masked sample: $\tilde{X} = M_k(X)$
> - Encoder: $p_{\theta}(\cdot)$
> - Original latent representation: $Z = p_{\theta}(X)$
> - Masked latent representation: $\tilde{Z} = p_{\theta}(\tilde{X})$
>
> **By the definition of mutual information.**
>
> $$
> I(X; Z)=H(Z)-H(Z\mid X),\qquad
> I(X; \tilde{Z})=H(\tilde{Z})-H(\tilde{Z}\mid X).
> $$
>
> **Under a deterministic encoder.**
>
> When the encoder parameters $\theta$ are fixed, $Z=p_{\theta}(X)$ and $\tilde{Z}=p_{\theta}(\tilde{X})$ are deterministic functions. Hence,
>
> $$
> H(Z\mid X)=0,\qquad H(\tilde{Z}\mid X)=0.
> $$
>
> Therefore
>
> $$
> I(X; Z)=H(Z),\qquad I(X; \tilde{Z})=H(\tilde{Z}).
> $$
>
> Furthermore, since $\tilde{X}$ is obtained from $X$ by masking, and
>
> $$
> Z=p_{\theta}(X),\qquad \tilde{Z}=p_{\theta}(\tilde{X}),
> $$
>
> assert that
>
> $$
> H(\tilde{Z}) < H(Z).
> $$
>
> Therefore
>
> $$
> I(X; \tilde{Z}) < I(X; Z).
> $$
>
> **Q4**: Is the ESP novel in the time-series literature - I didn't see any references on it.
>
> **A4**: Thank you gives us an opportunity to clarify the innovative contribution of the ESP method. The citations to prior work you pointed out are indeed important, and we will add a discussion of them in the Related Work section of the supplementary material.
>
> The core idea of ESP is to impose an input–output consistency constraint using relative distances: if two inputs x1,x2 are similar, then their corresponding outputs y1,y2 should also be similar. We note that the semi-supervised learning literature already contains consistency-regularization approaches with a related spirit. For example, the π-Model [1] applies two different perturbations (e.g., noise or data augmentation) to the same input and forces the model’s perturbed predictions to be consistent; Temporal Ensembling [1] constructs an unsupervised loss by taking a moving average of predictions across training epochs; and Mean Teacher [2] introduces a teacher model (often with stronger augmentations) as the consistency target for the student model in order to stabilize predictions.
>
> However, ESP differs fundamentally from these methods in its problem setting, object of constraint, and optimization objective. Consistency methods in semi-supervised learning primarily focus on the stability of a single sample’s predictions under different perturbations. By contrast, ESP is designed specifically for the unique challenges of time-series forecasting (e.g., semantic inconsistency and representation collapse): it explicitly aligns the geometric structure of the input embedding space with that of the output space to enhance robustness, while not relying on unlabeled data or data augmentation—making it simpler and more efficient to use. The novelty of ESP lies in introducing the idea of manifold alignment into time-series forecasting for the first time, directly constraining pairwise relative distances between samples rather than enforcing per-sample perturbation consistency, which better matches the characteristics of sequential data.
>
> [1] Laine S, Aila T. Temporal ensembling for semi-supervised learning[J]. arXiv preprint arXiv:1610.02242, 2016.
>
> [2] Tarvainen A, Valpola H. Mean teachers are better role models: Weight-averaged consistency targets improve semi-supervised deep learning results[J]. Advances in neural information processing systems, 2017, 30.
>
> **Q5**: The combination of AML and ESP is reminiscent of contrastive objectives in (self and semi-supervised) representation learning, have you explored connections?
>
> **A5**: Thank you for raising this insightful question! We have indeed thought deeply about the connections between AML/ESP and (self-/semi-)supervised representation learning, and we agree this is a very worthwhile direction to explore.
>
> Specifically, the design of ESP is inspired by the idea of consistency regularization in self-supervised learning—akin to positive-pair alignment in contrastive learning. ESP improves representation robustness by enforcing geometric consistency between the input and output spaces (i.e., $\Delta E \approx \Delta O$). Unlike standard contrastive methods, however, ESP does not rely on data augmentation or explicit negative sampling. Instead, it directly leverages the supervisory signal—the output-space distance $\Delta O$—to shape the geometry of input representations, which makes it particularly well suited to time-series forecasting.
>
> Similarly, the masking mechanism in AML does have parallels to semi-supervised learning: a masked input and its original counterpart naturally form a contrasting pair, and the model’s predictions for the masked variant can play a role analogous to pseudo-labels.

---

> ### Comment · Reviewer_EQP6 · 2025-08-03
>
> Thanks for providing clarifications and addressing my concerns, this further supports my score.
>
> Regarding Q3, the inequality should not be strict (without further assumptions/justification). That is,  $ H(\tilde{Z}) < H(Z)$ does not hold in general, $ H(\tilde{Z}) \leq H(Z)$ does. In particular, if the encoder is invariant to masking the equality will hold. I do not find the information theoretic interpretation/motivation entirely justified, although I acknowledge this line of argument is customary in e.g. representation learning, and I still think the paper makes valuable contributions.

---

> > ### Author Response · Authors · 2025-08-05
> > **Planned Revisions in the Manuscript**
> >
> > Thank you for your recognition of our work. The question you identified is indeed crucial and will materially strengthen the paper’s theoretical rigor. We will revise the main text to adopt a more rigorous formulation.

---

### Decision · Program_Chairs · 2025-09-17

**Decision:**

Accept (spotlight)

**Comment:**

This paper presents a novel approach to time series forecasting, Adaptive Masking Loss with Representation Consistency (AMRC), which aims to mitigate redundant feature learning by selectively masking temporal segments and enforcing representation consistency.  Across several datasets and backbones, AMRC yields consistent accuracy gains; ablations indicate complementarity between AML and ESP.

Strengths: This paper is based on an interesting empirical observation: masking/truncation can outperform full-context training. The proposed
AMRC enjoys simplicity and flexibility: AML and ESP are conceptually simple, architecture-agnostic, and plug-and-play. Empirical results have indicated improvements across multiple datasets and model families. Code is provided.

Weaknesses:  There are several limitations that may reduce the overall impact. The conceptual connections to information theory are not always clear, and the framing around the limits of long input sequences seems at odds with prior work, requiring more careful positioning. The paper lacks discussion of efficiency trade-offs and robustness to hyperparameter and masking choices, as well as clarity on what types of features AML tends to suppress. Some baseline models and datasets are missing, and the empirical claims about efficiency are not substantiated. Reviewers also note issues with writing consistency and insufficiently developed comparisons to related work.

Rebuttals: The authors have provided thorough responses to the concerns. Reviewers are confident with the potential values of this technical contribution.

Overall, this work proposes a promising and novel idea with solid empirical results, but it requires stronger theoretical grounding, clearer positioning against existing literature, and deeper analysis of efficiency and masking behavior to reach the bar of a top-tier venue.